# MiniCache: KV Cache Compression in Depth Dimension for Large Language Models

Akide Liu[1]    Jing Liu[1]    Zizheng Pan[1]    Yefei He[2]

Gholamreza Haffari[1]    Bohan Zhuang[1,2][†]

[1]ZIP Lab, Monash University, Australia
[2]ZIP Lab, Zhejiang University, China

## Abstract

A critical approach for efficiently deploying computationally demanding large language models (LLMs) is Key-Value (KV) caching. The KV cache stores key-value states of previously generated tokens, significantly reducing the need for repetitive computations and thereby lowering latency in autoregressive generation. However, the size of the KV cache grows linearly with sequence length, posing challenges for applications requiring long context input and extensive sequence generation. In this paper, we present a simple yet effective approach, called MiniCache, to compress the KV cache across layers from a novel depth perspective, significantly reducing the memory footprint for LLM inference. Our approach is based on the observation that KV cache states exhibit high similarity between the adjacent layers in the middle-to-deep portion of LLMs. To facilitate merging, we propose disentangling the states into the magnitude and direction components, interpolating the directions of the state vectors while preserving their lengths unchanged. Furthermore, we introduce a token retention strategy to keep highly distinct state pairs unmerged, thus preserving the information with minimal additional storage overhead. Our MiniCache is training-free and general, complementing existing KV cache compression strategies, such as quantization and sparsity. We conduct a comprehensive evaluation of MiniCache utilizing various models including LLaMA-2, LLaMA-3, Phi-3, Mistral, and Mixtral across multiple benchmarks, demonstrating its exceptional performance in achieving superior compression ratios and high throughput. On the ShareGPT dataset, LLaMA-2-7B with cross-layer merging achieves a compression ratio of $1.53\times$. Additionally, since MiniCache is orthogonal to existing quantization techniques, it can achieve a compression ratio of up to $5.02\times$ when combined with the 4-bit quantization technique, enhancing inference throughput by approximately $5\times$ and reducing the memory footprint by $41\%$ compared to the FP16 full cache baseline, all while maintaining near-lossless performance. Project is available at https://minicache.vmv.re .

## 1 Introduction

Large Language Models (LLMs), exemplified by the GPT series [1, 2, 3] and the LLaMA series [4, 5, 6], have emerged as pivotal innovations within the artificial general intelligence, significantly enhancing the capabilities of natural language processing. However, these models are meticulously trained using extensive computational resources [7] and massive datasets [8], which enables them to

---

[†]Corresponding author. Email: bohan.zhuang@gmail.com

38th Conference on Neural Information Processing Systems (NeurIPS 2024).

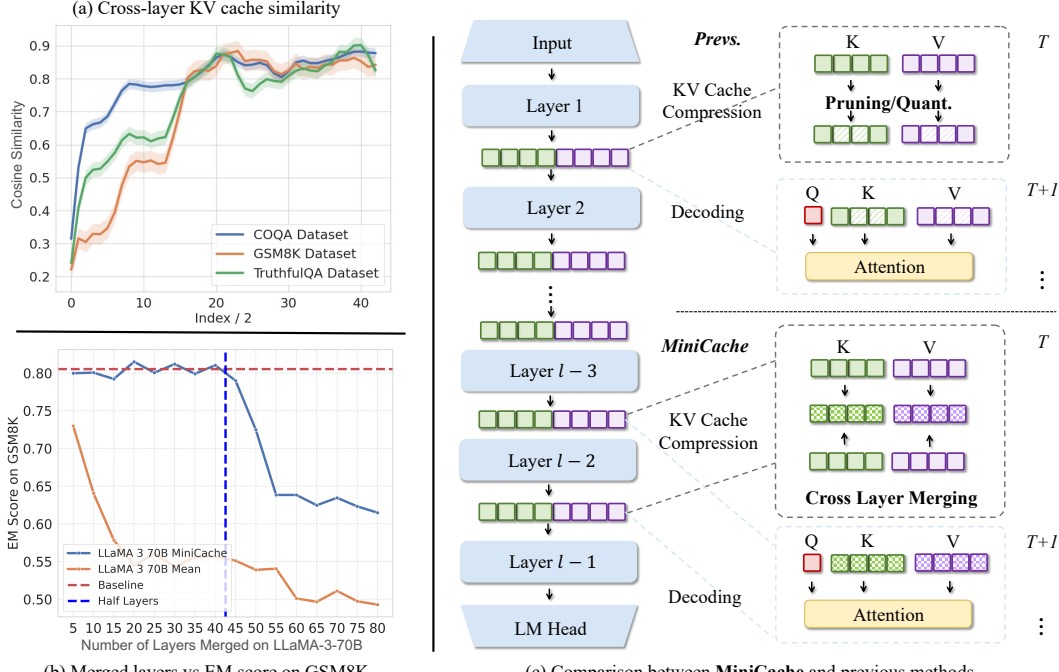

Figure 1: Overview of our MiniCache strategy and example results: (a) shows the observation that the KV cache states between two adjacent layers are highly similar, particularly across the middle to deep layers. The x-axis uses index/2 to represent the similarities for each pair of layers. (b) compares the performance of MiniCache, and the mean baseline, which simply averages the KV caches of two layers, using the LLaMA-3-70B model [6] on the GSM8K dataset [10]. MiniCache, which begins merging from the half-layer depth, achieves near-lossless performance. (c) highlights the primary difference between MiniCache and previous approaches. MiniCache investigates the inter-layer redundancy of KV caches along the depth dimension of LLMs, an aspect overlooked by intra-layer-based methods. Here, $T$ refers to the last timestamp of pre-filling, and $T + 1$ des to the first timestamp of decoding.

produce text that effectively mimics human writing styles and conducts complex reasoning analysis, yet raises the challenge of efficient deployment and serving. Within the inference framework of LLMs, KV caches [9] are crucial for storing pre-computed keys and values, thus avoiding repeated calculations over the preceding context and substantially enhancing LLMs' deployment efficiency. However, the increasing demand for longer sequence lengths results in voluminous cached states, leading to significant memory consumption during generation. For instance, a 175B GPT-3 model [2], with a batch size of 64 and a sequence length of 4,096 tokens (both prefilled and generated), requires approximately 1,208GB of GPU memory. This requirement is $3.45\times$ greater than the memory used to store the model's weights. In this context, KV cache compression is of paramount importance due to its clear benefits: 1) it largely reduces the memory footprint allowing for faster generation and larger batch serving; 2) it significantly lowers the cost per token, demonstrating substantial commercial benefits.

Existing KV cache compression efforts can be roughly categorized into two types, namely quantization and sparsity. The quantization approaches [11, 12] propose storing the KV states in low-bit numerical values. Typically, FlexGen [13] demonstrates that 4-bit KV cache quantization can achieve lossless performance. In contrast, sparsity-driven methods aim to retain only the salient tokens while evicting the rest, either heuristically [14, 15] or adaptively [16]. Some approaches [11] explore the intersection of these two types, by assigning high-bit to salient tokens and extremely low-bit to the rest of the tokens, achieving more aggressive memory gain. Despite these innovations, *existing literature merely consider the* **intra-layer redundancy**, *while neglecting another important complementary direction – the* **inter-layer redundancy**, as illustrated in the Figure 1(c).

Our analysis begins by exploring the redundancy of KV caches **along the depth dimension**, as shown in Figure 1(a). We observe that KV cache states exhibit high similarity between neighbouring layers in the middle-to-deep portions of LLMs. This intriguing property suggests that states paired by position between adjacent layers can be accurately merged into a single state space with a strong performance guarantee, as illustrated in Figure 1(b). This approach significantly reduces the memory footprint without the need to retain individual states for each attention layer. Note that these observations are pertinent to dynamic inference strategies such as mixture-of-depths [17] and layer-wise early exiting [18, 19], which optimize computational paths by skipping non-critical layers to enhance training and inference efficiency. Furthermore, layer pruning methods [20] highlight considerable redundancy in deeper layers. However, despite these advancements, the redundancy of KV caches along the depth dimension has largely been overlooked.

In this paper, we propose **MiniCache**, a simple yet effective cross-layer KV cache compression method aimed at advancing the inference efficiency of LLMs. MiniCache consists of two essential components. Firstly, we introduce an accurate cache merging strategy, employing a reparameterization of state vectors that decompose them into the magnitude and direction components, akin to weight normalization [21]. This approach allows for effective interpolation of the directional component in polar coordinates while preserving the original state norms to retain as much information as possible. This interpolation refers to the cross-layer merging as shown in the Figure 1(c). Secondly, we recognize that a small subset of state pairs, characterized by low similarities but carrying largely distinct semantic meanings, are unsuitable for inter-layer merging. To address this, we propose a token retention strategy to minimize performance degradation, which involves separately retaining these outlier pairs. Our framework is notably memory-efficient, requiring storage for only a single high-dimensional directional component, along with minimal extra memory overhead. The overhead consists of a few unmergeable tokens and their corresponding indexes, as well as token-wise magnitudes to accurately restore the original states.

We conduct extensive experiments with representative LLMs, including Mixtral-8x7B [22], Phi-3-Mini [23], and LLaMA-3 [6] 8B and 70B, respectively. Our method is benchmarked across a diverse range of question answering and generation datasets [24, 25, 26, 27, 28, 29, 30, 31] using the lm-eval-harness [32]. Additionally, we evaluate our results on LongBench [33] for long-sequence generation. The results demonstrate that MiniCache can reduce the memory footprint required for LLM inference by up to 41%, while simultaneously enhancing throughput by approximately $5\times$ compared to fully cached baseline, clearly surpassing existing methods [11, 12, 14, 15].

Our contributions are summarized as follows:

- We introduce MiniCache, a simple yet highly effective framework for KV cache compression. MiniCache pioneers the exploration of KV cache compression along the depth dimension, thereby significantly expanding its capabilities.

- We observe a fascinating characteristic of cross-layer KV cache states: high similarity between adjacent layers in the middle to later stages of LLMs. Additionally, we find that not all state pairs are suitable for merging.

- We propose an accurate and memory-efficient method for cross-layer cache merging, comprising a reparameterization strategy and a token retention mechanism. Our method complements existing KV cache compression approaches, further enhancing LLM serving efficiency.

- Our MiniCache performs favourably against the state-of-the-art methods. Notably, our 4-bit MiniCache achieves a strong compression ratio of up to $5.02\times$, $5\times$ higher inference throughput, and 41% memory reduction compared to the FP16 full cache baseline with near-lossless performance.

## 2 Related Work

**Efficient inference for LLMs.** Large Language Models (LLMs) are constrained by considerable computational and memory requirements during inference, particularly in resource-constrained environments. To mitigate these challenges, a variety of efficient inference techniques have been developed. For instance, dynamic inference methods [18, 34, 35, 36, 37, 38], represented by mixture-of-experts (MoE) [39, 40, 41, 42, 43], adaptively select specific sub-structures of the model during the inference process based on the input data, significantly improving inference efficiency while keeping model capacity. Techniques like Multi-Query Attention [44, 45], Kernel-driven attentions

[46, 47, 48, 49], and low-rank attentions [43, 50, 51, 52] approximate the functionality of traditional attention mechanisms but with more efficient implementations. Quantization strategies [53, 54, 55, 56] involve converting the model's weights and activations into a low bit-width format, thereby reducing memory footprint and computational intensity. Sparsification approaches [14, 15, 57, 58] eliminate unnecessary elements in both model weights and token representations, further enhancing efficiency. Some closely related works, such as MoD [17] and LayerSkips [19], considered the dynamic inference nature to ignore unimportant layers according to input [59] . However, these methods require an additional fine-tuning process or carefully designed pre-training stages, which reduces the adaptivity of these methods. MiniCache relies on inter-layer similarity observations to perform cross-layer merging, significantly reducing memory demand.

**Model merging.** Merging compression involves the aggregation of a model's parameters and activations at various granularities. This process enhances the efficiency of inference in large models and facilitates huge redundancy [60]. Linear Mode Connectivity (LMC) [61] enables the fine-tuning of models from a shared pre-trained base. Commonly, weight averaging [62] is employed as an efficient technique to perform merge compression. Notably, Model Soup [63] utilizes linear averaging in this context. Advanced methods like TIES Merging [64], Model Breadcrumbs [65], and DARE [66] further enhance this process by sparsifying and combining model parameters, enabling the merging of models without sacrificing performance capabilities. Additionally, Spherical Linear intERPolation (SLERP) [67] extends beyond simple weight averaging by interpolating between model parameters. The Fisher information matrix [68] and RegMean-based methods [69] further optimize merges to produce ideal weights, minimizing the $\ell_2$ distance to generation outputs while preserving the privacy of the training data. However, most existing works focus on merging model parameters, with the concept of depth-wise mergeability not being thoroughly explored in prior research. MiniCache focuses on the KV cache token merging in the depth dimensional of LLMs.

## 3 Motivation

In the below, we present our new observations in a novel cross-layer perspective.

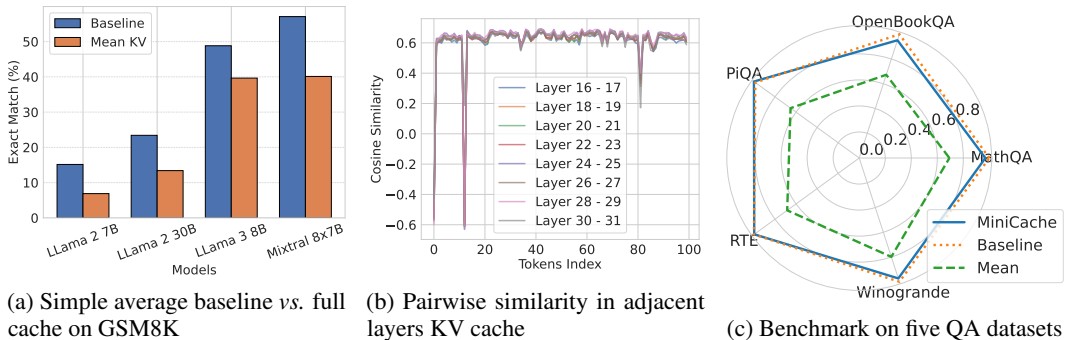

(a) Simple average baseline *vs.* full cache on GSM8K

(b) Pairwise similarity in adjacent layers KV cache

(c) Benchmark on five QA datasets

Figure 2: Overall of our explorations and observations : (a) shows the strong baseline by performing average merging on the KV cache. (b) shows the pairwise similarity of cache states between adjacent layers. (c) compares the MiniCache, simple average, and full cache baseline across five different datasets.

### 3.1 Cross-Layer Redundancy in KV Cache

Prior studies have revealed the ineffectiveness of middle-to-deep layers in LLMs [20]. Thus, layer-wise early exiting in this case can effectively avoid the redundant computation with minor effect on LLM performance [19, 70]. Inspired by this, we explore a *layer-wise* merging of KV cache in LLMs, starting with a simple baseline by averaging full tokens across adjacent layers. We provide our key observations as follows.

**Observation 1: KV cache shares a high similarity between adjacent layers.** Based on LLaMA-3-70B [6], we conduct zero-shot inference on the validation sets of three widely recognized benchmarks: COQA [71], GSM8K [10] and TruthfulQA [72]. In general, we find that KV cache in the shallow layers exhibits low similarity, whereas those in the middle-to-deep layers are very similar to one

another based on angular distance, as shown in Figure 1(a). Next, we merge the KV cache across adjacent layers by conducting five-shot inference with LLaMA-2-7B, LLaMA-2-13B [5], LLaMA-3-8B [6], and Mixtral-8x7B [22] on GSM8K [10]. Specifically, starting from the middle layer of each model, we merge the KV cache in the adjacent two layers. As shown in Figure 2(a), we observe a favourable performance for different LLMs, which reveals the huge potential for efficiency improvement by sharing the KV cache across adjacent layers during LLM decoding.

**Observation 2: Not all tokens are equally important to merge, a few distinct pairs require retention.** Recent works [15, 16] in KV cache compression have found that keeping a few salient tokens at each layer, which contribute the majority of attention scores, could be sufficient to maintain LLM performance. In our case, we speculate that certain token pairs in adjacent layers also exhibit outlier behaviours, showing strong semantic differences that make them unsuitable for merging. Based on COQA [71] and LLaMA-2-7B [5], we investigate the similarities at the level of token pairs. As shown in Figure 2(b), we find a significant portion of token pairs share high similarities across adjacent layers. However, we observe a few outlier pairs, such as indices 0 and 15, with large margins of difference. We consider these tokens non-mergeable due to their significant differences. We also show that merging these distinct tokens results in performance degradation, corresponding to $\gamma = 0$ row in Table 2. Thus, while cross-layer merging is a promising strategy for reducing memory burdens, it must be implemented with careful consideration of token-level similarities to ensure optimal performance, as shown in Figure 2(c).

# 4 Method

In this section, we introduce our MiniCache, a simple yet effective method aimed at trimming the KV cache redundancy in the depth dimension. This framework exploits the high similarity of KV cache states between adjacent layers and consists of two primary components: a reparameterization-based merging strategy and a token retention mechanism. The merging strategy compresses the KV cache states in adjacent layers to aggregate them into a single shared memory space, beginning from the middle of the model. The token retention mechanism mitigates information lost by retaining the highly distinct state pairs with minimal additional memory cost. With the merged cache, retention tokens, and magnitudes, we can accurately restore the original cache states for token decoding.

## 4.1 Cross-Layer Compression

Our method commences with the identification of an optimal starting layer $S$. Observations in Section 3.1 indicate that the KV cache from middle-to-deep layers consistently exhibits patterns of high similarity across adjacent layers. Consequently, we select the starting layer from the middle of the LLM, specifically $S = L/2$. From this layer onward, the KV pairs are assumed to be sufficiently similar across adjacent layers to warrant their consolidation. Central to this approach is a merge function, $F$, which is designed to integrate the KV caches of consecutive layers into a single, unified cache. We define $x$ as the vectorized cache state of a single token, where the superscript indicates the layer index and the subscripts $k$ and $v$ denote the keys and values, respectively. Specifically, for a pair of key/value tokens at the same position in layers $l$ and $l-1$, the merged cache is computed as

$$
\begin{aligned}
\boldsymbol{c}_k^{l,l-1} &= F(\boldsymbol{x}_k^l, \boldsymbol{x}_k^{l-1}), \\
\boldsymbol{c}_v^{l,l-1} &= F(\boldsymbol{x}_v^l, \boldsymbol{x}_v^{l-1}).
\end{aligned}
\tag{1}
$$

This consolidation process effectively eliminates the need to store and process the original memory-intensive keys and values in each layer independently. Instead, it approximates a shared cache across the adjacent layers.

## 4.2 KV Cache Merging and Restoration

**Reparameterization-based cache merging.** To perform the pairwise merging, one solution is to directly average a pair of KV tokens, analogous to model merging [63, 64]. However, we observe that direct averaging can cause significant information loss. We conjecture that the distance between activations can be larger than that of weights due to the presence of outlier activation channels with extremely large magnitudes in LLMs [73, 74], while weights typically have relatively quite small magnitudes. A potential method to compensate for this information loss is to project from $\boldsymbol{c}$ to $\boldsymbol{x}^{l-1}$

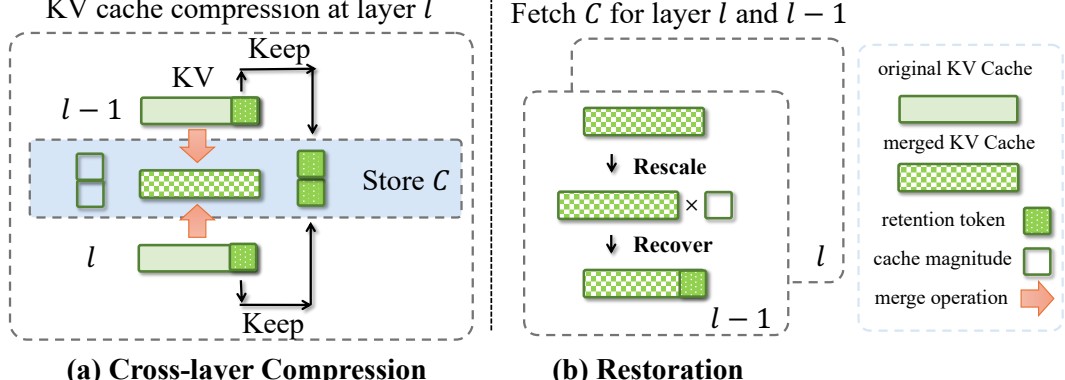

| KV cache compression at layer $l$ | Fetch $C$ for layer $l$ and $l-1$ | |

**(a) Cross-layer Compression**  **(b) Restoration**

Figure 3: The illustration of the proposed method **MiniCache**. (a) depicts the cross-layer compression process. We fetch the KV caches, from layers $l$ and $l-1$, and merge them into shared states via Eq. (3). Additionally, we compute the $\ell_2$ norm for the caches to obtain their magnitudes. Furthermore, we select unmergable tokens for retention, then store merged cache, retention tokens, and magnitudes at layer $l$ in $C$. (b) illustrates the restoration process for layers $l$ and $l-1$, which includes magnitude rescaling in Eq. (2) and retention token recovery.

and $\boldsymbol{x}^l$, then rescale the projected vectors based on their relative magnitudes to exactly restore the original states. However, this approach requires extensive additional storage and computations; for example, restoring $\boldsymbol{x}^{l-1}$ needs both $\boldsymbol{c}$ and $\boldsymbol{x}^l$, which undermines the benefits of cache merging. To efficiently merge token pairs, we draw inspiration from weight normalization [21], which disentangles model parameters into the *magnitude* and *direction* components to accelerate the convergence of stochastic gradient descent. Additionally, we take cues from DoRA [75], which employs a similar way to resemble the learning behavior of parameter-efficient fine-tuning compared to full fine-tuning. In our case, the reparameterization can be formulated as follows:

$$\hat{\boldsymbol{x}}^l = \boldsymbol{e}^{l,l-1} \cdot \frac{\|\boldsymbol{x}^l\|}{\|\boldsymbol{e}^{l,l-1}\|}, \ \hat{\boldsymbol{x}}^{l-1} = \boldsymbol{e}^{l,l-1} \cdot \frac{\|\boldsymbol{x}^{l-1}\|}{\|\boldsymbol{e}^{l,l-1}\|}, \tag{2}$$

where $\boldsymbol{e}$ is the directional vector. This decomposition ensures that $\frac{\boldsymbol{e}^{l,l-1}}{\|\boldsymbol{e}^{l,l-1}\|}$ is a unit vector, and allows the restored states to match the $\ell_2$ norm of the original states, thereby preserving the cache's information as much as possible. The restoration is shown as Figure 3(b). For brevity, we omit the subscripts $k$ and $v$, as keys and values are decomposed in the same way. For estimating the directional component $\boldsymbol{e}^{l,l-1}$, we follow SLERP [67], which adaptively handles the interpolation, which often resembles rotation-like transformations. The choice of SLERP as the merging function is strategic, as it facilitates interpolation along the shortest path on the unit sphere between two high-dimensional vectors, thereby preserving their geometric integrity, which refers to merge operation in Figure 3(a). This is crucial for maintaining the semantic and syntactic properties of the KV caches. The formula for SLERP in our context is:

$$\boldsymbol{e}^{l,l-1} = \frac{\sin((1-t)\Omega^{l,l-1})}{\sin(\Omega^{l,l-1})} \cdot \frac{\boldsymbol{x}^{l-1}}{\|\boldsymbol{x}^{l-1}\|} + \frac{\sin(t\Omega^{l,l-1})}{\sin(\Omega^{l,l-1})} \cdot \frac{\boldsymbol{x}^l}{\|\boldsymbol{x}^l\|}, \tag{3}$$

where $\Omega^{l,l-1} = \arccos\left(\frac{\boldsymbol{x}^l \cdot \boldsymbol{x}^{l-1}}{\|\boldsymbol{x}^l\|\|\boldsymbol{x}^{l-1}\|}\right)$ represents the angle between vectors $\boldsymbol{x}^l$ and $\boldsymbol{x}^{l-1}$, and $\sin(\cdot)$ is the sine function. $t$ is an interpolation hyperparameter that adjusts the relative influence of each vector on the resulting direction, tailored to the layer depth and specific characteristics of the KV pairs. Note that when we set $t = 0.5$, it will become an average merging along the geometry surface, which we consider special cases in Eq. (A). The merged cache for each token pair is then a concatenation of the directional vector, magnitude and $\Omega^{l,l-1}$, denoting as $\boldsymbol{c}^{l,l-1} = [\boldsymbol{e}^{l,l-1}, \|\boldsymbol{x}^{l-1}\|, \|\boldsymbol{x}^l\|, \Omega^{l,l-1}]$, cached components as shown in Figure 3(a). Note that apart from storing the merged directional vector, we only need to store additional token-wise magnitude and angle scalars, which is memory efficient. In this way, we achieve substantial memory efficiencies through reduced redundancy while ensuring the retention of the critical functional characteristics of the original KV pairs across transformer layers.

**Unmergeable token retention.** Highly distinct pairs are sensitive to merging operations, leading us to propose unmergeable token retention, as shown in Figure 3(a). Despite the high similarity between

KV cache states across neighbouring layers, a few sensitive distinct pairs remain that are significantly difficult to merge and share. Aligned with previous studies, these distinct tokens carry substantial semantic meanings [15, 16]. We observe that merging sensitive tokens, which results in the loss of layer-specific information, can lead to significant performance degradation. Therefore, it is crucial to properly disentangle the shared and unique information between adjacent layers. To address this issue, we designed a token retention strategy to selectively retain tokens that cannot be merged based on their angular distance, defined as: $d(\boldsymbol{x}^l, \boldsymbol{x}^{l-1}) = \frac{1}{\pi}\Omega$. For the KV caches, the minimum and maximum angular distances are determined to identify the unmergeable tokens.

The set of required token indices to keep, $\mathbb{I}$, is obtained by:

$$\mathbb{I} = \{i \mid d_i < d_{\min} + (d_{\max} - d_{\min}) \cdot \gamma\}, \tag{4}$$

where $\gamma$ is a predefined hyperparameter that controls the retention threshold. The tokens with indices in $\mathbb{I}$ are retained and not compressed during the merge, which ensures that performance does not decline by preventing the loss of unmergeable tokens.

Next, let $\boldsymbol{X} \in \mathbb{R}^{n \times h}$ be either the key or value cache at one attention layer, where $n$ denotes the number of tokens and $h$ is the number of hidden dimensions, and $\boldsymbol{E} \in \mathbb{R}^{n \times h}$ be the shared KV cache states. For each pair of neighbouring two layers, the unmergeable tokens are selected along with the token dimension by $\boldsymbol{R}^l = \boldsymbol{X}^l[\mathbb{I}]$, $\boldsymbol{R}^{l-1} = \boldsymbol{X}^{l-1}[\mathbb{I}]$, then restoring to our compressed caches by $\hat{\boldsymbol{X}}^l[\mathbb{I}] = \boldsymbol{R}^l$, $\hat{\boldsymbol{X}}^{l-1}[\mathbb{I}] = \boldsymbol{R}^{l-1}$, as shown in Figure 3(b). Overall, we share the final cache for the two layers as $\boldsymbol{C}^{l,l-1} = [\boldsymbol{E}^{l,l-1}, \boldsymbol{R}^l, \boldsymbol{R}^{l-1}, \|\boldsymbol{X}^{l-1}\|, \|\boldsymbol{X}^l\|, \mathbb{I}]$. This cache includes the shared KV cache states, retention of unmerged tokens, magnitude vectors for each layer, and the token-keeping index, respectively. These additional components are quite lightweight. Thus compared to full-layer caches, our method remains memory-efficient, as discussed in Sec. 4.3.

**Cache restoration.** After obtaining the shared cache $\boldsymbol{C}^{l,l-1}$, we further need to approximately restore the original cache states for the current token decoding, as shown in Fig. 3(b). Specifically, to restore $\boldsymbol{X}^l$, we first rescale the directional shared states with the corresponding magnitude vector along the token dimension, denoted as $\boldsymbol{E}^{l,l-1}\|\boldsymbol{X}^l\|$. Subsequently, we perform retention token recovery by placing the sensitive tokens according to their token indices.

### 4.3 Efficiency Discussion

**Compression efficiency.** We primarily analyze our memory efficiency in terms of the number of tokens used. Next, let $r$ be the number of layers and and $b$ is the batch size, $s$ and $n$ are input and output sequence length respectively. We consider the FP16 storage for KV cache. The full cache memory usage is given by $4brh(s+n)$. In our study, we begin merging layers from the middle to the deeper layers, consolidating the KV cache states of every two layers into a single shared state space. As a result, we effectively reduce the GPU memory usage in the decoding inference to $3brh(s+n)$, demonstrating a significant compression rate.

**Restoration efficiency.** We then analyze the additional memory cost incurred during the restoration process, which During the magnitude rescaling phase, we save an additional norm vector for the corresponding layers in the KV cache. It is important to note that the norm vector is in the shape of $\mathbb{R}^{b \times s \times 1}$, which has a single channel dimension compared to the fully ranked original KV states. Additionally, we suppose that the retention threshold can be set to 0.05. Therefore, we have $brh(0.05(s+n))$ tokens retained without compression. Finally, our overall memory requirement is given by $(3.1h+2)br(s+n)$. The detailed derivation is shown in the Appendix E.

## 5 Experiments

We demonstrated that our MiniCache can perform merging compression on the latter half of the layers of LLMs with minimal performance degradation.

**Implementation details.** Our experiments are based on representative model families of LLMs, including a compact LLM Phi-3-Mini [23] and an MoE LLM Mixtral-8x7B [22]. Additionally, we adopt LLaMA-3 [6] 8B and 70B models to explore how our method generalizes to larger LLMs. We sample ten tasks from lm-eval-harness [32], including COPA [24], MathQA [25], OpenBookQA [26], PIQA [27], RTE [28], WinoGrande [29], XSUM [30], and CNN/Daily Mail [31]. We also evaluate long-sequence generation on LongBench [33]. We compare our method with a fully cached baseline,

and other methods such as round-to-nearest quantization (RTN) [76], SmoothQuant [73] and KIVI [11].

For the proposed MiniCache, we set the interpolation parameter $t$ to 0.6, indicating that the merged results have a smaller rotation angle to the next layer. Furthermore, we set the token retention threshold $\gamma$ to 0.05, according to the statistics of unmergeable tokens across multiple datasets. In addition to our merging method, we also consider a strong baseline of average merging. For sequential loading of large models, we utilize NVIDIA 4 A100 80GB GPUs, more details refers to Appendix D.

**Main results.** We evaluate MiniCache by merging KV caches across all layers on GSM8K, COQA, and TruthfulQA. The results are shown in Figure 4. In general, we demonstrate the general effectiveness of merging KV caches from middle-to-deep layers across different sized LLMs. Moreover, the proposed MiniCache demonstrates a consistent and significant advantage over the averaging baseline. We also illustrate the performance of merging KV caches across half of the layers with the blue lines, where MiniCache still maintains a robust performance and achieves the best compression ratio. Besides, we find that our method is even more effective for larger LLMs. For instance, based on LLaMA-3-70B, MiniCache shows nearly zero performance drop even with the KV cache in 87.5% of the layers merged on the COQA dataset. This highlights the adaptability and efficiency of our approach in handling large-scale models while ensuring minimal performance degradation.

**LongBench.** We also conduct experiments to evaluate performance and quality in long-sequence generation using the LongBench dataset [33], as shown in Table 1. Our experiments applied MiniCache over several models: LLaMA-2-7B-Chat, LLaMA-2-13B-Chat, Mistral-7B, and Mistral-7B-Instruct. It is important to note that our MiniCache method maintains orthogonality with all existing quantization and sparsity (refers to Table A) methods at both the model and token-wise levels. When combined with KIVI-4bit KV cache quantization, our approach achieves a compression rate of $5.02\times$, with minimal impact on accuracy across various challenging long-context generation tasks. The combination of MiniCache and KIVI-4bit KV cache quantization demonstrates significant memory savings without compromising the model's ability to handle long sequences effectively. This high-

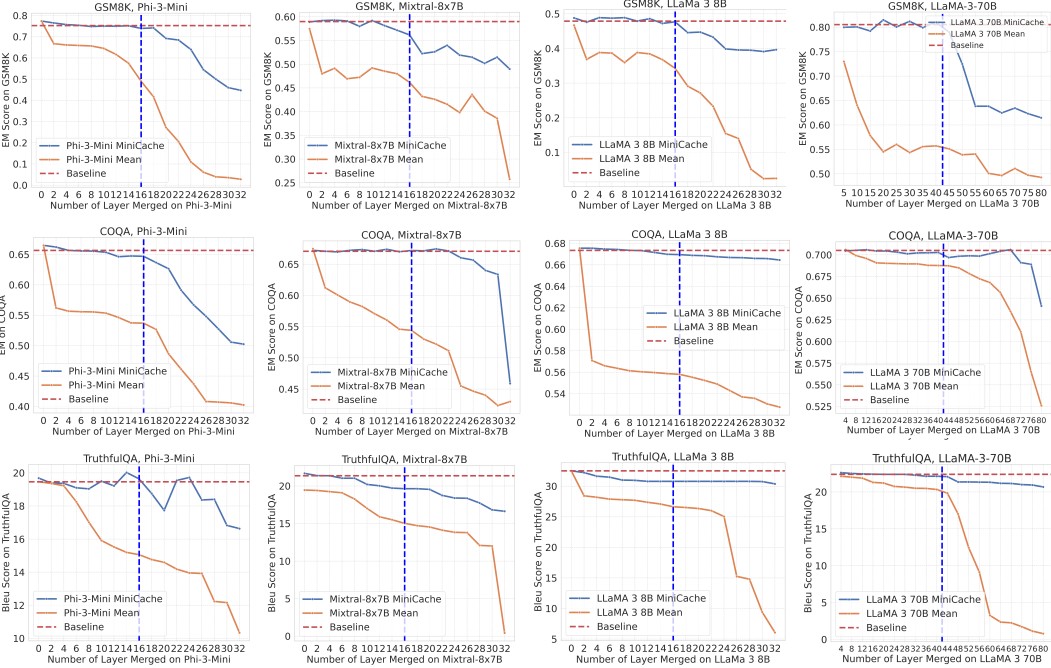

Figure 4: Performance comparisons between our proposed MiniCache with the "averaging baseline" and the "unmerged full cache baseline" on multiple datasets with Phi3-Mini, Mixtral-8x7B, LLaMA-3-8B, and LLaMA-3-70B. More result details are shown in Appendix F. The x-axis indicates the number of layers merged. As more layers are merged, a greater reduction in memory usage is achieved.

Table 1: Evaluation of different KV cache compression methods on LongBench. MiniCache builds on top of 4-bit KIVI [11] and achieves the best performance with the strongest compression rate.

| Model | Method | LCC | RepoBench-P | PR-en | TREC | 2wikimqa | GovReport | MQA-zh | Average | Compression Ratio |
|---|---|---|---|---|---|---|---|---|---|---|
| Llama-2-7B-Chat | Baseline | 58.16 | 52.19 | 10.12 | 64.00 | 31.12 | 27.09 | 10.12 | 36.41 | 1x |
| | RTN [76] | 15.44 | 8.76 | 0.79 | 4.00 | 0.30 | 1.93 | 0.07 | 4.90 | 3.21x |
| | SmoothQuant [73] | 35.31 | 32.18 | 0.79 | 28.75 | 7.45 | 11.83 | 1.68 | 16.28 | 2.15x |
| | KIVI-2 [11] | 49.32 | 43.71 | 4.50 | 63.00 | 24.07 | 24.73 | 10.24 | 31.51 | 3.95x |
| | **MiniCache** | 58.03 | 52.01 | 9.00 | 64.00 | 30.58 | 25.32 | 10.13 | **35.44** | **5.02x** |
| Llama-2-13B-Chat | Baseline | 48.06 | 50.08 | 14.25 | 68.50 | 13.09 | 27.76 | 7.23 | 32.71 | 1x |
| | RTN [76] | 20.89 | 18.62 | 0.33 | 0.00 | 0.52 | 1.68 | 0.16 | 6.03 | 3.21x |
| | SmoothQuant [73] | 32.17 | 33.86 | 2.65 | 48.00 | 3.53 | 12.47 | 0.47 | 19.16 | 2.15x |
| | KIVI-2 [11] | 48.60 | 48.81 | 13.50 | 68.00 | 14.32 | 25.70 | 7.01 | 32.42 | 3.95x |
| | **MiniCache** | 48.75 | 48.59 | 13.00 | 68.00 | 14.36 | 26.57 | 7.99 | **32.61** | **5.02x** |
| Mistral-7B | Baseline | 68.06 | 60.46 | 17.71 | 68.00 | 10.87 | 20.09 | 17.10 | 37.33 | 1x |
| | RTN [76] | 27.98 | 26.18 | 3.34 | 13.00 | 1.11 | 2.49 | 0.45 | 10.51 | 3.21x |
| | SmoothQuant [73] | 40.63 | 35.14 | 3.40 | 30.50 | 6.03 | 5.00 | 4.12 | 17.55 | 2.15x |
| | KIVI-2 [11] | 65.16 | 58.33 | 12.43 | 65.00 | 11.03 | 13.22 | 13.87 | 33.43 | 3.95x |
| | **MiniCache** | 68.89 | 60.98 | 13.92 | 67.00 | 10.50 | 18.06 | 7.88 | **35.75** | **5.02x** |
| Mistral-7B-Instruct | Baseline | 55.51 | 48.96 | 60.00 | 71.00 | 27.33 | 32.85 | 42.74 | 48.32 | 1x |
| | RTN [76] | 32.36 | 33.23 | 0.67 | 1.00 | 2.25 | 10.03 | 2.30 | 11.55 | 3.21x |
| | SmoothQuant [73] | 43.84 | 38.63 | 4.79 | 39.50 | 10.34 | 23.61 | 8.33 | 24.43 | 2.15x |
| | KIVI-2 [11] | 53.13 | 48.60 | 47.50 | 69.00 | 20.68 | 29.37 | 33.88 | 43.74 | 3.95x |
| | **MiniCache** | 54.79 | 51.02 | 64.14 | 71.00 | 24.97 | 31.46 | 27.54 | **46.99** | **5.02x** |

lights the potential of our method to optimize large language models for tasks requiring extensive context, making them more efficient and scalable for real-world applications.

**Efficiency analysis.** To assess the acceleration capabilities of MiniCache, we conduct evaluations based on the methodologies employed in vLLM [77] and KIVI [11]. We generate synthetic workloads derived from ShareGPT, which include real input and output texts from LLM services. The dataset features an average input prompt length of 161 tokens and an output length of 338 tokens. Using the LLaMA-2-7B model on a single 80GB NVIDIA A100 GPU, we benchmark our method in a batch-serving scenario, comparing peak memory usage and throughput among 2-bit KIVI, 4-bit MiniCache, and an FP16 baseline. As illustrated in Figure 5, with a batch size of 128, **MiniCache reduces memory usage by 25GB, achieving a 41% memory saving**. In terms of throughput, **MiniCache outperforms the FP16 baseline by approximately 5×**. Additionally, despite utilizing 4-bit quantization, MiniCache benefits from merging and sharing KV caches across adjacent layers, resulting in a 1.29× higher throughput compared to the 2-bit KIVI. These results demonstrate that MiniCache offers a state-of-the-art trade-off between efficiency and performance.

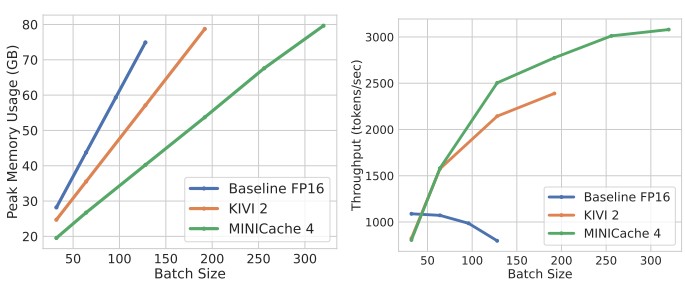

(a) BS. *vs.* Peak Memory Usage  (b) BS. *vs.* Decoding Throughput

Figure 5: Memory usage and throughput comparison between our 4-bit MiniCache, 2-bit KIVI, and 16-bit Baseline. MiniCache can achieve higher throughput by enabling a larger batch size while reducing memory footprints via LLaMA-2-7B [5].

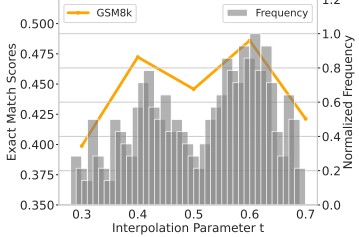

Figure 6: LLaMA-3-8B [6] to experiment on the GSM8K [10]. The right axis is the normalized frequency of the relative magnitude ratio. Optional $t$ shows a strong correlation with frequency.

## 6 Ablation Study

**The effect of interpretation parameter $t$.** We explore the effects of the interpretation parameter $t$ on performance, particularly in relation to the relative magnitude ratio of adjacent layers, as shown in Figure 6. We maintain all settings constant, starting from layer $S = 16$ (halfway

Table 2: Comparisons of various token retention thresholds $\gamma$ by LLaMA-2-7B [5] on three benchmarks.

| $\gamma$ | COQA | GSM8K | TruthfulQA |
|---|---|---|---|
| 0 | 0.603 | 0.108 | 29.813 |
| 0.01 | 0.620 | 0.126 | 30.226 |
| 0.02 | 0.630 | 0.143 | 33.903 |
| 0.05 | 0.647 | 0.152 | 33.213 |
| 0.1 | 0.643 | 0.152 | 33.903 |
| 1 | 0.643 | 0.159 | 33.743 |

through the layers of LLaMA-3-8B), and vary the interpretation parameter $t$ from 0.3 to 0.7. Our findings reveal several key points. When $t = 0.5$, the process resembles average merging, which is less effective for cross-layer merging. In contrast, when $t = 0.6$ is optimal, the merged representation exhibits the most robust performance, while indicating that more information is derived from the second term ($\boldsymbol{x}^l$) of the SLERP.

The frequency results also indicate that the high frequencies are clustered around 0.4 and 0.6, corroborating our optimal $t$. Moreover, there is a strong correlation between the optimal $t$ and the high frequency of the relative magnitude ratio of adjacent layers. This finding provides an opportunity to utilize the relative magnitude ratio to dynamically determine the interpretation parameter $t$. Dynamic $t$ allows for more flexible weight control in SLERP merging for each layer-wise operation, thereby showing potential for further exploration.

**The effect of token retention threshold $\gamma$.** We investigate the impact of the token retention threshold $\gamma$ on model performance across the three datasets, as shown in Table 2. A larger $t$ generally means retaining more tokens for improved performance, but this comes at the cost of increased memory demand. The results suggest that setting $\gamma$ to 0.05 achieves the best balance between performance and efficiency.

# 7 Conclusion and Future Work

This paper presents a pioneering exploration of KV cache compression in the depth dimension, addressing a significant memory bottleneck in LLMs. Our proposed MiniCache offers a simple, effective, and training-free approach to compressing KV caches by leveraging the notable high similarities between KV caches in neighboring layers, starting from the midpoint of LLMs. We have demonstrated that MiniCache can significantly reduce the memory footprint required for LLM inference by up to 41%, while simultaneously enhancing throughput by approximately five times compared to the FP16 baseline. In conclusion, MiniCache significantly advances the field of KV cache compression, offering a state-of-the-art balance between efficiency and performance. Future work will focus on enhancing the compression ratio by cross-multiple-layer merging, developing advanced merging algorithms such as Spherical Cubic Interpolation [78], and further optimizing memory usage for large-scale deployments in diverse application scenarios.

# 8 Acknowledgement

This research is partially supported by the ARC Future Fellowship (FT190100039) to G.H. Additional support was partially provided by the DARPA Assured Neuro-Symbolic Learning and Reasoning (ANSR) program under award number FA8750-23-2-1016.

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

# Appendix

## A  Additional Experiment Results

**Comparisons with token sparsity methods**. We also compare MiniCache with the sparsity-based method H2O [14]. Our method outperforms H2O in most tasks on LongBench. Notably, our approach focuses on reducing inter-layer redundancy, whereas H2O focuses on intra-layer redundancy. Our approaches are orthogonal to sparsity-based methods.

Table A: Comparison between MiniCache with token sparsity based method H2O, using mistral-7B-instruct on LongBench dataset.

| Methods | NrtvQA | Qasper | MF-en | HotpotQA | 2WikiMQA | Musique | GovReport | QMSum | MultiNews | TREC | TriviaQA | SAMSum | PCount | PRe | Lcc |
|---|---|---|---|---|---|---|---|---|---|---|---|---|---|---|---|
| Baseline | 26.82 | 33.06 | 49.28 | 42.77 | 27.33 | 19.27 | 32.85 | 24.25 | 27.06 | 71.0 | 86.23 | 42.98 | 2.75 | 86.98 | 55.51 |
| H2O[14] | 22.61 | 29.06 | 47.22 | 36.54 | 20.6 | 16.25 | 30.0 | 23.8 | 26.75 | 70.5 | 86.16 | 42.97 | 3.46 | 86.38 | 53.72 |
| Attention Sink [79] | 24.67 | 31.26 | 45.74 | 41.86 | 22.17 | 17.64 | 29.15 | 21.40 | 25.17 | 69.66 | 85.94 | 40.60 | 2.94 | 81.19 | 51.74 |
| MiniCache | 27.04 | 32.59 | 49.38 | 43.91 | 24.97 | 18.3 | 31.46 | 23.85 | 26.64 | 71.0 | 86.93 | 43.6 | 3.04 | 79.56 | 54.79 |

**Execution time comparison for difference sequence length.** We benchmark the latency of LlaMA-2-7B on an NVIDIA A100 GPU using different sequence lengths ranging from 1024 to 4096 with a batch size of 16, as shown in table B. We compare it with H2O, which requires calculating full attention scores to estimate token importance. In contrast, MiniCache performs reparameterization-based merging and token restoration using simple matrix manipulations, resulting in more lightweight computations and lower latency. Specifically, when the sequence length is 4096, MiniCache shows a 36.83% reduction in latency compared to H2O.

Table B: Execution time comparison for different sequence lengths. We benchmarked the latency of LLaMA-2-7B on an NVIDIA A100 GPU using sequence lengths ranging from 1024 to 4096 with a batch size of 16.

| | 1024 | 2048 | 3072 | 4096 |
|---|---|---|---|---|
| H2O | 9083.08 ms | 19,274.46 ms | 26,508.06 ms | 45,850.708 ms |
| **MiniCache** | 8605.53 ms | 14,843.40 ms | 23,111.93 ms | 33,414.56 ms |

**Trade-off between accuracy and efficiency.** We conducted experiments by varying the ratio of retention tokens, as shown in table C. We observe that accuracy improves up to a certain point before plateauing as the token retention ratio increases. Retaining 20% of the tokens is necessary to ensure all salient tokens are preserved without performance degradation. In contrast, we only need to retain the top 5% of the salient tokens using a dynamic distance-based threshold, as proposed in our paper. Note that efficiency decreases as more tokens are retained. This demonstrates that our distance-based approach better balances performance and efficiency than the fixed retention ratio counterpart.

**Ablation for relations between performance and ratio of merged tokens.** We are adding another column in terms of compression ratio to represent the efficiency trade-offs, as shown in table D. The results suggest that setting $\gamma$ to 0.05 achieves the best balance between performance and efficiency.

Table C: Performance comparison of LLaMA-2-7B under different retention ratios of overall tokens.

| n% token | COQA | GSM8K | TruthfulQA |
|---|---|---|---|
| 0 | 0.603 | 0.108 | 29.813 |
| 0.01 | 0.607 | 0.114 | 29.997 |
| 0.05 | 0.614 | 0.137 | 30.174 |
| 0.1 | **0.625** | **0.139** | **32.661** |
| 0.2 | 0.641 | 0.152 | 33.174 |
| 1 | 0.643 | 0.159 | 33.743 |

Table D: Performance and compression ratios of LLaMA-2-7B at various retention thresholds $\gamma$ in Eq. (4).

| $\gamma$ | COQA | GSM8K | TruthfulQA | Compression Ratio |
|---|---|---|---|---|
| 0 | 0.603 | 0.108 | 29.813 | 5.191 |
| 0.01 | 0.620 | 0.126 | 30.226 | 5.156 |
| 0.02 | 0.630 | 0.143 | 33.903 | 5.122 |
| 0.05 | **0.647** | **0.152** | **33.213** | **5.023** |
| 0.1 | 0.643 | 0.152 | 33.903 | 4.866 |
| 1 | 0.643 | 0.159 | 33.743 | 3.115 |

**Benchmark of the computational overhead of different components.** As shown in Table table E, the reparametrization process, including the computation of magnitude and direction, takes 0.093 ms (0.031 ms + 0.062 ms), and the restoration process, including the computation of the distance matrix and token replacement, takes 0.116 ms (0.061 ms + 0.055 ms). These times indicate a negligible computational overhead compared to the overall attention computation, which takes 9.756 ms. The running time is measured in milliseconds per attention layer with a batch size of 1 on LLaMA-2-7B.

Table E: We benchmark the different components in the reparameterization and restoration stages using LLaMA-2-7B.

| Component | Running Time (ms) | Standard Variation (ms) |
|---|---|---|
| Magnitude | 0.031 | 0.007 |
| Direction | 0.062 | 0.013 |
| Distance | 0.061 | 0.022 |
| Token Replacement | 0.055 | 0.010 |
| Attention (Overall) | 9.756 | 0.513 |

**Efficiency comparison without quantization.** table F demonstrates that MiniCache's cross-layer merging without quantization achieves almost lossless performance compared to the FP16 baseline, with a compression ratio of 1.53. In contrast, 2-bit quantization causes performance degradation. For instance, on the GSM8K dataset, performance drops from 0.159 to 0.127, but the compression ratio improves to 3.95. Our MiniCache method, focusing on the depth dimension, can complement any quantization and existing KV cache compression methods. The results indicate that combining cross-layer merging with 4-bit quantization achieves the optimal balance between performance and efficiency.

Table F: Performance comparison and compression ratios across different methods. MiniCache is orthogonal to existing quantization techniques.

| Method | COQA | GSM8K | TruthfulQA | Compression Ratio |
|---|---|---|---|---|
| FP16 | 0.643 | 0.159 | 33.743 | 1 |
| KIVI-2 | 0.631 | 0.127 | 33.950 | 3.95 |
| KIVI-4 | 0.638 | 0.154 | 33.864 | 3.29 |
| Cross-layer Merging | 0.644 | 0.159 | 33.713 | 1.53 |
| **Cross-layer Merging + 4-bit quantization** | **0.647** | **0.152** | **33.213** | **5.023** |

# B   Additional Related Work

**Preliminaries of KV cache in LLMs.** LLM inference has been significantly improved in recent works [9, 13, 77] by optimizing the KV cache management. Overall, this line of research is typically done in two steps: 1) First, at the *prefilling* stage, LLM calculates the initial KV caches at each attention layer based on the input sequence and decodes the first token. Second, 2) at the *decoding* stage, LLM autoregressively predicts the next token, where its KV cache at each attention layer is added to the overall caches. Existing works have compressed KV cache in different aspects (e.g., quantization [11, 12], token pruning [14, 16] ).

**KV cache compression.** In the prior study, various strategies for enhancing efficient transformer architectures are discussed, covering a spectrum of techniques aimed at optimizing performance and managing resource constraints. These methods include attention optimization [48, 49, 80], grouping queries [44, 45], sparse KV caching [16, 81, 82], shrinking tokens [15, 83], and improving long-context generation. Significant contributions come from projects such as H2O [15], GEAR [15], and KIVI [11]. Additionally, efforts to alleviate KV cache bottlenecks include strategies like multi-query attention [44] and multi-group attention [45], which propose reducing the number of heads in the KV cache. However, these methods often necessitate retraining or fine-tuning models. Other approaches focus on diminishing the size of the KV cache by selectively evicting less important

tokens [14] and enhancing the system architecture through technologies like offloading the KV cache [84] or integrating techniques such as virtual memory and paging [85] into the attention mechanism.

## C   Discussions and Limitations

Table G: Comparison of performance using different cross-layer merging strategies. The experiment shows that SLERP has the best performance across three datasets.

| Strategy | COQA | GSM8K | TruthfulQA |
|---|---|---|---|
| Average | 0.541 | 0.081 | 28.162 |
| Max Norm | 0.607 | 0.104 | 30.150 |
| SLERP | **0.647** | **0.152** | **33.213** |

**Alternative merging function.** During our preliminary exploration, we initially considered an alternative, simpler merge function for cross-layer compression: maximum norm-preserving interpolation. This function is designed to maintain the maximum norm of the vectors involved, ensuring that the most significant features are preserved during the merging process. The maximum norm-preserving interpolation in terms of $F_{\max}$ can be defined as follows:

$$F_{\max}(\boldsymbol{x}^l, \boldsymbol{x}^{l-1}) = \frac{\bar{\boldsymbol{x}}^{l,l-1}}{\|\bar{\boldsymbol{x}}^{l,l-1}\|} \cdot \mathrm{Max}(\|\boldsymbol{x}^l\|, \|\boldsymbol{x}^{l-1}\|). \tag{A}$$

Here $\bar{\boldsymbol{x}}^{l,l-1}$ represents the average vector between $\boldsymbol{x}^l$ and $\boldsymbol{x}^{l-1}$. The function $F_{\max}$ ensures that the merged vector preserves the direction of the average vector while scaling it to the maximum norm of the original KV states. Compared to the SLERP-based merge function, $F_{\max}$ has less computational overhead and lower memory consumption. However, it is less accurate than SLERP. The choice between using $F_{\mathrm{SLERP}}$ or $F_{\max}$ depends on the specific requirements of the application. In our study, we primarily use SLERP to maximize performance, as also shown in Table G.

**Societal impact.** Our work shows a preliminary exploration of KV cache Compression in the depth dimension, a relatively unexplored yet critically bottlenecked area in large language models (LLMs). The proposed MiniCache provides a solution to improve the efficiency of LLM generation and is adaptable to existing intra-layer-based KV cache pruning and quantization technologies. Additionally, we proposed a simple yet effective approach that merges similar KV cache states in a cross-layer manner and effectively restores performance through a novel restoration technique. Our observations indicate that cross-layer similarity and mergeability can be applied in broad-efficient applications for post-training optimization in low-resource scenarios, such as deployment on mobile devices. Moreover, with effective optimization of KV cache memory demand, MiniCache further enhances long-context generation, which is a crucial paradigm for real-world applications, such as understanding concepts in textbooks. We aim for our work to advance the boundaries of two key challenges in the LLM industry and research: batch inference and long-context generation.

Furthermore, in addition to the significant contributions of MiniCache for KV cache Compression, several challenges remain that are common to LLMs. Issues such as the truthfulness and security of LLMs are still unresolved. Ensuring the accuracy and reliability of generated content is critical, as LLMs can sometimes produce plausible but incorrect or misleading information. Additionally, safeguarding against security vulnerabilities, such as adversarial attacks or data leakage, is paramount to maintaining the integrity and confidentiality of user interactions. Addressing these challenges requires ongoing research and development to enhance the robustness and trustworthiness of LLMs. This effort must proceed alongside advancements in computational efficiency and performance, as exemplified by innovations like MiniCache.

**Limitations.** The current merging algorithm based on Spherical Linear Interpolation (SLERP) has its limitations. SLERP is suitable only for merging two vectors and uses an interpolation approach that restricts our algorithm from merging multiple layers simultaneously and maximizing the compression ratio in further states. This limitation impacts the overall efficiency of KV cache compression and underscores the need for advanced techniques capable of handling more complex merging scenarios.

Future research should focus on developing more sophisticated algorithms that can overcome these constraints, thereby enhancing the compression capabilities and overall performance of LLMs.

# D   Additional Implementation Details

**Overview of the inference algorithm.** The MiniCache inference implementation, as shown in Alg. 1, involves several key steps. When the interface starts, In the prefilling phase, we define the merging starting layer $S$. Before reaching layer $S$, the inference uses the original attention and cache logic. From layer $S$ onward, we implement our merging algorithm, which operates in a cross-layer manner and is applied only at odd-numbered layers. During merging, we fetch the KV cache from the previous layer and save the merged shared states into the current layer's KV cache. To reduce memory usage, we then remove the cache of the previous layer. Additionally, we store the magnitudes vector and retention-sensitive tokens for each layer. During the decoding phase, two scenarios arise after layer $S$. For even-numbered layers (first round), since the KV cache has been removed during the prefilling phase, we refer to the next layer $(l + 1)$ to fetch the shared KV cache states. We then perform approximated scale restoration and retention token recovery. The new KV states from this phase are stored for use in the next round. In the second round, which involves odd-numbered layers, we use the new KV tokens from both the previous and current layers. After the restoration phase, we perform the merge operations and update the shared KV cache states in the stack.

**Cross-Layer merging and restoration algorithm.** As outlined in Alg. 2, MiniCache algorithm involves several crucial steps to ensure efficient memory usage while maintaining the integrity of the Key-Value (KV) Cache states. Initially, given the KV cache $\boldsymbol{E}_{k,v}^{l}$, norm values $\|\boldsymbol{X}_{k,v}^{l}\|$ unmerged tokens $\boldsymbol{R}_{k,v}^{l}$, retention indices $\mathbb{I}_{k,v}$, and the next tokens $\boldsymbol{t}^{l}$, $\boldsymbol{t}^{l-1}$, the algorithm proceeds by rescaling the magnitude of the KV pairs. Specifically, $\hat{\boldsymbol{X}}_{k}^{l}$ and $\hat{\boldsymbol{X}}_{v}^{l}$ are computed by multiplying the normalized KV pairs $\boldsymbol{E}_{k,v}^{l}$ with their respective magnitude norms $\|\boldsymbol{X}_{k,v}^{l}\|$. Following this, the algorithm restores unmerged tokens using the retention indices, updating $\hat{\boldsymbol{X}}_{k}^{l}$ and $\hat{\boldsymbol{X}}_{v}^{l}$ accordingly. Next, the new tokens $\boldsymbol{t}_{k}$ and $\boldsymbol{t}_{v}$ are concatenated to the rescaled KV pairs along the token dimension. This augmented KV cache undergoes a softmax attention mechanism where the attention scores $\boldsymbol{A}$ are computed by taking the dot product of the query token $\boldsymbol{t}_{q}$ with the transposed keys $(\hat{\boldsymbol{X}}_{k}^{l})^{\top}$. The output token $\boldsymbol{t}_{O}$ is then obtained by multiplying the attention scores $\boldsymbol{A}$ with the values $\hat{\boldsymbol{X}}_{v}^{l}$. In cases where the previous token $\boldsymbol{t}^{l-1}$ exists, the algorithm performs a compression step. It concatenates the existing KV cache $\boldsymbol{E}_{k,v}^{l}$ with the merged tokens resulting from the current and previous layers, effectively reducing redundancy and optimizing memory. If $\boldsymbol{t}^{l-1}$ is not available, the KV cache is updated by simply concatenating $\boldsymbol{E}_{k,v}^{l}$ with the new tokens $\boldsymbol{t}_{k,v}^{l}$, deferring the compression until the next iteration. The final output token $\boldsymbol{t}_{O}$ is then returned, concluding the decoding process. In the merging function, the algorithm normalizes the KV pairs from both the current and previous layers. It calculates the angular distance $\Omega$ between the normalized vectors, ensuring that the interpolation occurs along the shortest path on the unit sphere. The merged KV cache is then obtained by weighted interpolation of the normalized vectors, preserving the geometric and semantic integrity of the original states. This comprehensive process allows MiniCache to achieve substantial memory efficiencies while maintaining the functional characteristics of the KV pairs across transformer layers.

**MiniCache execution flow.** Figure A delineates the pre-filling and decoding logic for the MiniCache framework, which incorporates cross-layer merging and error suppression to achieve memory efficiency and maintain functional integrity. Initially, in Step 1, the KV cache is fetched from the previous layer (Layer $L-1$) during the pre-filling phase. In Step 2, the fetched KV pairs from the current layer $\chi^{L}$ are merged with the KV pairs from the preceding layer $\chi^{L-1}$, reducing redundancy through a merge function. Subsequently, in Step 3, the merged KV pairs are cached for future use, representing a consolidated data set from multiple layers. During the decoding phase, Step 4 involves deleting unnecessary or redundant KV pairs to optimize memory usage. In Step 5, the decoder fetches the required KV pairs from the cache for output generation. Step 6 applies error suppression mechanisms to the fetched KV pairs, including rescaling and retention recovery, to minimize errors introduced during the merging and compression processes. Finally, in Step 7, the cache is updated with the final KV pairs post-error suppression and adjustments, ensuring the cache contains the most accurate and efficient representation of the KV pairs for subsequent layers. This comprehensive approach guarantees substantial memory efficiencies while preserving the critical functional characteristics of the original KV pairs across transformer layers.

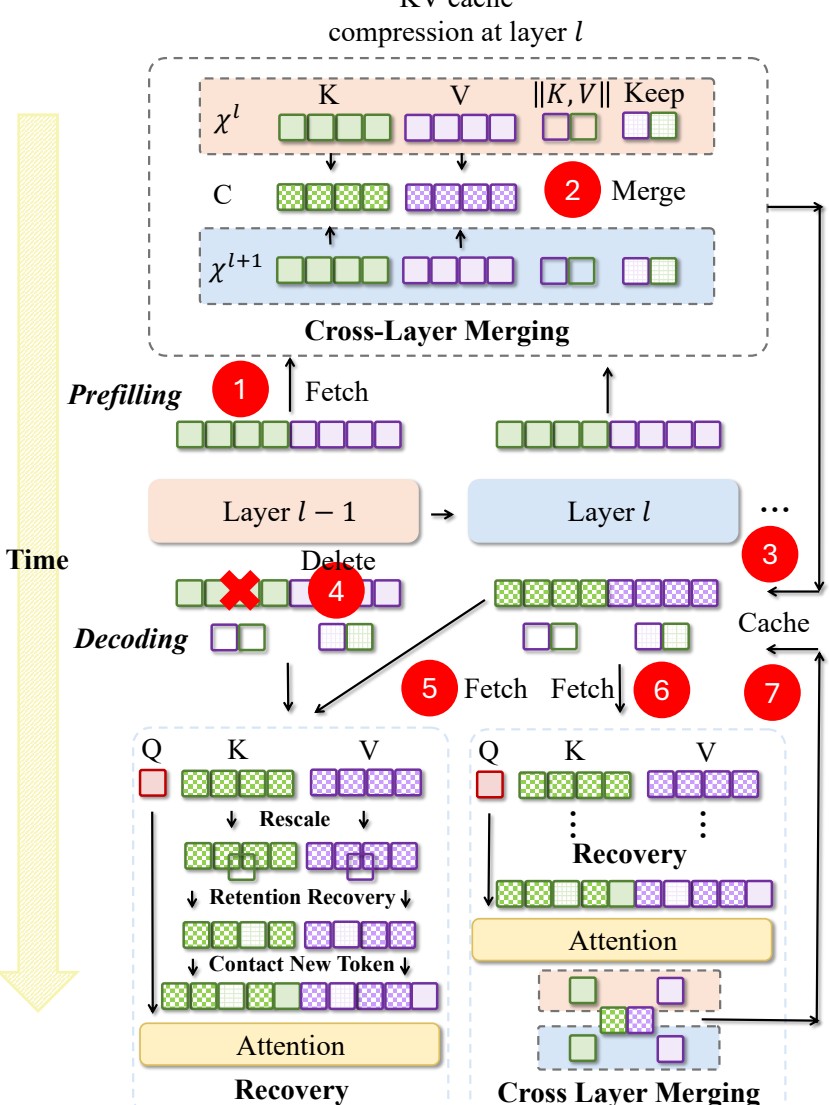

Figure A: Overall prefilling and decoding logic for **MiniCache** involves performing cross-layer merging and recovery within our framework.

## E  Detailed Efficiency Derivation

In this section, we provide a detailed derivation of the memory efficiency improvements outlined in Section 4.3.

First, we consider the original KV cache memory usage, which is given by:

$$4brh(s + n).$$

Here, $r$ is the number of layers, $b$ is the batch size, $h$ is the hidden size, $s$ is the input sequence length, and $n$ is the output sequence length. To improve efficiency, we begin merging layers starting from the midpoint, $S = \frac{1}{2}r$, by consolidating the KV cache states of every two layers into a single shared state space. The memory usage derivation proceeds as follows: for the unmerged part of the cache (from layer 1 to $S$):

**Algorithm 1:** The MiniCache Inference Algorithm

---

1   **procedure** MiniCache Inference:

     **Input:** Input Tokens: $\boldsymbol{T} \in \mathbb{R}^{t_{\text{input}} \times d}$, number of layers $L$, merging beginning layer $S$

     **Output:** Output Tokens: $\boldsymbol{O} \in \mathbb{R}^{t_{\text{output}} \times d}$

2     **for** $l \leftarrow 0$ **to** $S - 1$ **do**

3        **procedure** Standard Prefill:

4           Standard Attention & Standard Cache

5        **procedure** Standard Decoding:

6           Standard Attention & Standard Cache

     // Start Merging from layer $S$

7     **for** $l \leftarrow S$ **to** $L$ **do**

       // Perform Merging in every two layers $l\%2 == 1$

8        **if** $l\%2 == 1$ *and prefilling* **then**

9          **procedure** MiniCache Prefill:

            **Input:** KV cache from Current layer $l$: $\boldsymbol{X}_{k,v}^{l}$, KV cache from Previous layer

                $l - 1$: $\boldsymbol{X}_{k,v}^{l-1}$, token retention threshold: $\gamma$

10          **Delete** KV cache of the $l - 1$-th layer // layer $l, l-1$ shares one Cache

11        Standard Attention & Standard Cache

       // Perform Merging in the second layer

12        **if** *and decoding* **then**

13          **if** $l\%2 == 0$ **then**

            // first round in the cross-layer merging, fetch shared KV

               cache states from $\boldsymbol{C}_{k,v}^{l+1}$

14             **procedure** MiniCache Decoding:

               **Input:** KV cache: $\boldsymbol{C}_{k,v}^{l+1}$, Norm: $\|\boldsymbol{X}_{k,v}^{l}\|$, Unmerged Tokens: $\boldsymbol{R}_{k,v}^{l}$,

                  Retention indices: $\mathbb{I}_{k,v}$, Next Token: $t^{l}$

15          **else**

            // second round in cross-layer merging, while $t^{l-1}$ exist

16             **procedure** MiniCache Decoding:

               **Input:** KV cache: $\boldsymbol{C}_{k,v}^{l}$, Norm: $\|\boldsymbol{X}_{k,v}^{l}\|$, Unmerged Tokens: $\boldsymbol{R}_{k,v}^{l}$, Retention

                  indices: $\mathbb{I}_{k,v}$, Next Token: $t^{l}, t^{l-1}$

17     **return** $O$

---

$$4brh(s + n) \cdot \frac{1}{2} = 2brh(s + n).$$

For the merged part of the cache (from layer $S + 1$ to $r$):

$$4brh(s + n) \cdot \frac{1}{2} \cdot \frac{1}{2} = brh(s + n).$$

Combining these two parts, the total memory usage is:

$$2brh(s + n) + brh(s + n) = 3brh(s + n).$$

Next, we consider the additional memory cost incurred during the restoration process. During this phase, we save additional normalized vectors for the layers in the KV cache. These vectors are of shape $\mathbb{R}^{b \times s \times 1}$, which means they have a single channel dimension compared to the fully ranked original KV states.

The additional normalized vectors for layers from $S$ onwards are given by:

---

**Algorithm 2:** The MiniCache Prefill & Decoding Compression Algorithm

---

**Hyperparameter:** number of layers $L$, merging beginning layer $S$

1 **procedure** `MiniCache Prefill`:

    **Input:** KV cache from current layer $l$: $\boldsymbol{X}_{k,v}^{l} \in \mathbb{R}^{2 \times t_{\text{prompt}} \times d}$, KV cache from previous layer

        $l-1$: $\boldsymbol{X}_{k,v}^{l-1} \in \mathbb{R}^{2 \times t_{\text{prompt}} \times d}$, token retention threshold: $\gamma$

2    $\boldsymbol{E}_k^l, \|\boldsymbol{X}_k^l\|, \|\boldsymbol{X}_k^{l-1}\|, \Omega_k \leftarrow \text{Merge}(\boldsymbol{X}_k^l, \boldsymbol{X}_k^{l-1})$ ;

3    $\boldsymbol{E}_v^l, \|\boldsymbol{X}_v^l\|, \|\boldsymbol{X}_v^{l-1}\|, \Omega_v \leftarrow \text{Merge}(\boldsymbol{X}_v^l, \boldsymbol{X}_v^{l-1})$ ;

4    $\underbrace{\boldsymbol{E}_{k,v}^l \in \mathbb{R}^{t_{\text{prompt}} \times d}}_{\text{compression output}}, \underbrace{\|\boldsymbol{X}_{k,v}^{l,l-1}\| \in \mathbb{R}^{4 \times t_{\text{prompt}} \times 1}}_{\text{norms for rescaling}}$ ;

5    $d(\boldsymbol{X}^l, \boldsymbol{X}^{l-1})_{k,v} = \frac{1}{\pi} \cdot \Omega_{k,v}$ // `distance metrics`

6    $\mathbb{I}_{k,v} = \{i \mid d_i < d_{\min} + (d_{\max} - d_{\min}) \cdot \gamma\}$ // `retention indices`

7    $\boldsymbol{R}_k^{l,l-1} \leftarrow \boldsymbol{X}_k^{l,l-1}[\mathbb{I}_k], \boldsymbol{R}_v^{l,l-1} \leftarrow \boldsymbol{X}_v^{l,l-1}[\mathbb{I}_v]$ // `unmerged tokens`

8    **return** $\boldsymbol{E}_{k,v}^l, \|\boldsymbol{X}_{k,v}^{l,l-1}\|, \boldsymbol{R}_k^{l,l-1}, \boldsymbol{R}_v^{l,l-1}, \mathbb{I}_{k,v}$

9 **procedure** `MiniCache Decoding`:

    **Input:** KV cache: $\boldsymbol{E}_{k,v}^l \in \mathbb{R}^{2 \times t_{\text{prompt}} \times d}$, Norm: $\|\boldsymbol{X}_{k,v}^l\| \in \mathbb{R}^{2 \times t_{\text{prompt}} \times 1}$, Unmerged Tokens:

        $\boldsymbol{R}_{k,v}^l \in \mathbb{R}^{2 \times \gamma \cdot t_{\text{prompt}} \times d}$, Retention indices: $\mathbb{I}_{k,v} \in \mathbb{R}^{2 \times \gamma \cdot t_{\text{prompt}} \times 1}$, Next Token:

        $\boldsymbol{t}^l \in \mathbb{R}^{1 \times d}, \boldsymbol{t}^{l-1} \in \mathbb{R}^{1 \times d}$

10   $\hat{\boldsymbol{X}}_k^l \leftarrow \boldsymbol{E}_k^l \cdot \frac{\|\boldsymbol{X}_k^l\|}{\|\boldsymbol{E}_k^l\|} \; \hat{\boldsymbol{X}}_v^l \leftarrow \boldsymbol{E}_v^l \cdot \frac{\|\boldsymbol{X}_v^l\|}{\|\boldsymbol{E}_v^l\|}$ // `magnitude rescale`

11   $\hat{\boldsymbol{X}}_k^l[\mathbb{I}_k] = \boldsymbol{R}_k^l \; \hat{\boldsymbol{X}}_v^l[\mathbb{I}_v] = \boldsymbol{R}_v^l$ // `token restoration`

12   $\hat{\boldsymbol{X}}_k^l \leftarrow \text{Concat}(\hat{\boldsymbol{X}}_k^l, \boldsymbol{t}_k, \text{dim=token}) \; \hat{\boldsymbol{X}}_v^l \leftarrow \text{Concat}(\hat{\boldsymbol{X}}_v^l, \boldsymbol{t}_v, \text{dim=token})$

     $\boldsymbol{A} \leftarrow \text{Softmax}(\boldsymbol{t}_q \cdot (\hat{\boldsymbol{X}}_k^l)^\top) \; \boldsymbol{t}_O \leftarrow \boldsymbol{A} \cdot \hat{\boldsymbol{X}}_v^l$ **if** $\boldsymbol{t}^{l-1}$ *exists* **then**

13     | KV cache $\leftarrow \text{Concat}(\boldsymbol{E}_{k,v}^l, \text{Merge}(\boldsymbol{t}_{k,v}^l, \boldsymbol{t}_{k,v}^{l-1}), \text{dim=token})$ // `perform compression`

14   **else**

15     | KV cache $\leftarrow \text{Concat}(\boldsymbol{E}_{k,v}^l, \boldsymbol{t}_{k,v}^l, \text{dim=token})$ // `wait for compression`

16   **return** $\boldsymbol{t}_O$

17 **function** `MiniCache Merge`($\boldsymbol{X}^l, \boldsymbol{X}^{l-1}, t$):

18   $\vec{\boldsymbol{X}}^l \leftarrow \frac{\boldsymbol{X}^l}{\|\boldsymbol{X}^l\|}$

19   $\vec{\boldsymbol{X}}^{l-1} \leftarrow \frac{\boldsymbol{X}^{l-1}}{\|\boldsymbol{X}^{l-1}\|}$

20   $\Omega \leftarrow \arccos\left(\frac{\boldsymbol{X}_T^l \cdot \boldsymbol{X}_T^{l-1}}{\|\boldsymbol{X}_T^l\|\|\boldsymbol{X}_T^{l-1}\|}\right)$

21   $\boldsymbol{E} \leftarrow \frac{\sin((1-t)\Omega)}{\sin(\Omega)}\vec{\boldsymbol{X}}^l + \frac{\sin(t\Omega)}{\sin(\Omega)}\vec{\boldsymbol{X}}^{l-1}$

22   **return** $\boldsymbol{E}, \|\boldsymbol{X}^l\|, \|\boldsymbol{X}^{l-1}\|, \Omega$

---

$$br(s + n) \cdot 2 = 2br(s + n).$$

We also introduce a retention threshold, which we set to 0.05. This means that 5% of the KV cache tokens are retained without compression:

$$brh(0.05(s + n)).$$

Combining these terms, the total additional memory for the restoration process is:

$$2br(s + n) + 0.1brh(s + n).$$

Finally, summing the compressed memory usage and the restoration memory cost, the overall memory requirement is:

$$3brh(s + n) + 2br(s + n) + 0.1brh(s + n).$$

This can be simplified by grouping the common factors:

$$br(s + n)\left(3h + 2 + 0.1h\right).$$

Simplifying the expression inside the parentheses, we get:

$$br(s + n)\left(3.1h + 2\right).$$

Therefore, the total memory cost for the KV cache in the MiniCache Framework is:

$$br(s + n)(3.1h + 2).$$

This detailed derivation confirms the efficiency improvements discussed in Section 4.3, highlighting the significant reduction in memory usage achieved through our layer merging and restoration strategies.

# F  Detailed Experiment Results

Table H: Detailed performance comparison on GSM8K dataset with LLaMA-3-70B.

| Layer | LLaMA-3-70B MiniCache | LLaMA-3-70B Mean |
|---|---|---|
| 0 | 0.799 | 0.729 |
| 5 | 0.800 | 0.640 |
| 10 | 0.792 | 0.578 |
| 15 | 0.815 | 0.545 |
| 20 | 0.801 | 0.560 |
| 25 | 0.812 | 0.544 |
| 30 | 0.799 | 0.556 |
| 35 | 0.810 | 0.557 |
| 40 | 0.790 | 0.551 |
| 45 | 0.725 | 0.539 |
| 50 | 0.638 | 0.541 |
| 55 | 0.638 | 0.501 |
| 60 | 0.625 | 0.497 |
| 65 | 0.635 | 0.511 |
| 70 | 0.623 | 0.497 |
| 75 | 0.615 | 0.493 |

Table I: Detailed performance comparison on COQA dataset with LLaMA-3-70B.

| Layer | LLaMA-3-70B MiniCache | LLaMA-3-70B Mean |
|---|---|---|
| 0 | 0.705 | 0.706 |
| 4 | 0.705 | 0.699 |
| 8 | 0.706 | 0.696 |
| 12 | 0.704 | 0.691 |
| 16 | 0.704 | 0.690 |
| 20 | 0.703 | 0.690 |
| 24 | 0.701 | 0.690 |
| 28 | 0.702 | 0.690 |
| 32 | 0.702 | 0.688 |
| 36 | 0.703 | 0.688 |
| 40 | 0.697 | 0.687 |
| 44 | 0.698 | 0.685 |
| 48 | 0.699 | 0.678 |
| 52 | 0.699 | 0.672 |
| 56 | 0.701 | 0.668 |
| 60 | 0.704 | 0.657 |
| 64 | 0.706 | 0.635 |
| 68 | 0.691 | 0.611 |
| 72 | 0.689 | 0.565 |
| 76 | 0.641 | 0.526 |

Table J: Detailed performance comparison on TruthfulQA dataset with LLaMA-3-70B.

| Layer | LLaMA-3-70B MiniCache | LLaMA-3-70B Mean |
|---|---|---|
| 0 | 22.615 | 22.130 |
| 4 | 22.512 | 22.005 |
| 8 | 22.451 | 21.876 |
| 12 | 22.413 | 21.303 |
| 16 | 22.387 | 21.209 |
| 20 | 22.387 | 20.752 |
| 24 | 22.387 | 20.657 |
| 28 | 22.276 | 20.501 |
| 32 | 22.130 | 20.479 |
| 36 | 22.130 | 20.335 |
| 40 | 22.073 | 19.834 |
| 44 | 21.356 | 17.024 |
| 48 | 21.356 | 12.440 |
| 52 | 21.333 | 9.127 |
| 56 | 21.316 | 3.255 |
| 60 | 21.172 | 2.349 |
| 64 | 21.153 | 2.250 |
| 68 | 21.002 | 1.721 |
| 72 | 20.940 | 1.119 |
| 76 | 20.683 | 0.784 |

Table K: Detailed performance comparison on GSM8K dataset with LLaMA-3-8B.

| Layer | LLaMA-3-8B MiniCache | LLaMA-3-8B Mean |
|-------|----------------------|-----------------|
| 0     | 0.488                | 0.467           |
| 2     | 0.476                | 0.369           |
| 4     | 0.489                | 0.388           |
| 6     | 0.487                | 0.387           |
| 8     | 0.489                | 0.359           |
| 10    | 0.479                | 0.388           |
| 12    | 0.486                | 0.384           |
| 14    | 0.472                | 0.368           |
| 16    | 0.477                | 0.343           |
| 18    | 0.446                | 0.291           |
| 20    | 0.447                | 0.271           |
| 22    | 0.433                | 0.234           |
| 24    | 0.399                | 0.155           |
| 26    | 0.396                | 0.140           |
| 28    | 0.395                | 0.052           |
| 30    | 0.391                | 0.024           |
| 32    | 0.397                | 0.025           |

Table L: Detailed performance comparison on COQA dataset with LLaMA-3-8B.

| Layer | LLaMA-3-8B MiniCache | LLaMA-3-8B Mean |
|-------|----------------------|-----------------|
| 0     | 0.676                | 0.676           |
| 2     | 0.676                | 0.571           |
| 4     | 0.675                | 0.566           |
| 6     | 0.674                | 0.564           |
| 8     | 0.674                | 0.561           |
| 10    | 0.673                | 0.560           |
| 12    | 0.672                | 0.560           |
| 14    | 0.670                | 0.559           |
| 16    | 0.670                | 0.558           |
| 18    | 0.669                | 0.555           |
| 20    | 0.669                | 0.552           |
| 22    | 0.668                | 0.549           |
| 24    | 0.667                | 0.543           |
| 26    | 0.667                | 0.537           |
| 28    | 0.666                | 0.536           |
| 30    | 0.666                | 0.531           |
| 32    | 0.665                | 0.528           |

Table M: Detailed performance comparison on TruthfulQA dataset with LLaMA-3-8B.

| Layer | LLaMA-3-8B MiniCache | LLaMA-3-8B Mean |
|---|---|---|
| 0 | 32.520 | 32.524 |
| 2 | 32.231 | 28.431 |
| 4 | 31.645 | 28.197 |
| 6 | 31.485 | 27.894 |
| 8 | 31.008 | 27.796 |
| 10 | 30.964 | 27.704 |
| 12 | 30.798 | 27.371 |
| 14 | 30.798 | 27.093 |
| 16 | 30.798 | 26.643 |
| 18 | 30.798 | 26.517 |
| 20 | 30.798 | 26.355 |
| 22 | 30.798 | 26.011 |
| 24 | 30.798 | 25.044 |
| 26 | 30.798 | 15.254 |
| 28 | 30.798 | 14.791 |
| 30 | 30.765 | 9.419 |
| 32 | 30.390 | 6.068 |

Table N: Detailed performance comparison on GSM8K dataset with Mixtral-8x7B.

| Layer | Mixtral-8x7B MiniCache | Mixtral-8x7B Mean |
|---|---|---|
| 0 | 0.589 | 0.575 |
| 2 | 0.592 | 0.480 |
| 4 | 0.593 | 0.491 |
| 6 | 0.591 | 0.469 |
| 8 | 0.580 | 0.472 |
| 10 | 0.592 | 0.492 |
| 12 | 0.582 | 0.485 |
| 14 | 0.572 | 0.480 |
| 16 | 0.562 | 0.462 |
| 18 | 0.522 | 0.432 |
| 20 | 0.526 | 0.426 |
| 22 | 0.540 | 0.416 |
| 24 | 0.519 | 0.398 |
| 26 | 0.515 | 0.436 |
| 28 | 0.502 | 0.401 |
| 30 | 0.515 | 0.386 |
| 32 | 0.490 | 0.258 |

Table O: Detailed performance comparison on COQA dataset with Mixtral-8x7B.

| Layer | Mixtral-8x7B MiniCache | Mixtral-8x7B Mean |
|---|---|---|
| 0 | 0.672 | 0.675 |
| 2 | 0.671 | 0.612 |
| 4 | 0.670 | 0.601 |
| 6 | 0.672 | 0.590 |
| 8 | 0.674 | 0.582 |
| 10 | 0.671 | 0.571 |
| 12 | 0.674 | 0.561 |
| 14 | 0.670 | 0.546 |
| 16 | 0.672 | 0.544 |
| 18 | 0.672 | 0.530 |
| 20 | 0.675 | 0.522 |
| 22 | 0.671 | 0.512 |
| 24 | 0.660 | 0.455 |
| 26 | 0.657 | 0.447 |
| 28 | 0.640 | 0.440 |
| 30 | 0.634 | 0.424 |
| 32 | 0.459 | 0.430 |

Table P: Detailed performance comparison on TruthfulQA dataset with Mixtral-8x7B.

| Layer | Mixtral-8x7B MiniCache | Mixtral-8x7B Mean |
|---|---|---|
| 0 | 21.686 | 19.465 |
| 2 | 21.385 | 19.405 |
| 4 | 21.368 | 19.251 |
| 6 | 21.038 | 19.094 |
| 8 | 21.038 | 18.265 |
| 10 | 20.216 | 17.019 |
| 12 | 20.026 | 15.902 |
| 14 | 19.723 | 15.505 |
| 16 | 19.641 | 15.028 |
| 18 | 19.641 | 14.723 |
| 20 | 19.546 | 14.543 |
| 22 | 18.756 | 14.122 |
| 24 | 18.402 | 13.834 |
| 26 | 18.366 | 13.789 |
| 28 | 17.738 | 12.091 |
| 30 | 16.827 | 12.008 |
| 32 | 16.635 | 0.430 |

Table Q: Detailed performance comparison on GSM8K dataset with Phi-3-Mini.

| Layer | Phi-3-Mini MiniCache | Phi-3-Mini Mean |
|---|---|---|
| 0 | 0.774 | 0.774 |
| 2 | 0.765 | 0.667 |
| 4 | 0.757 | 0.661 |
| 6 | 0.754 | 0.659 |
| 8 | 0.748 | 0.657 |
| 10 | 0.750 | 0.645 |
| 12 | 0.750 | 0.616 |
| 14 | 0.752 | 0.575 |
| 16 | 0.739 | 0.491 |
| 18 | 0.742 | 0.417 |
| 20 | 0.692 | 0.272 |
| 22 | 0.685 | 0.206 |
| 24 | 0.640 | 0.110 |
| 26 | 0.545 | 0.061 |
| 28 | 0.500 | 0.039 |
| 30 | 0.460 | 0.036 |
| 32 | 0.447 | 0.028 |

Table R: Detailed performance comparison on COQA dataset with Phi-3-Mini.

| Layer | Phi-3-Mini MiniCache | Phi-3-Mini Mean |
|---|---|---|
| 0 | 0.665 | 0.665 |
| 2 | 0.662 | 0.562 |
| 4 | 0.657 | 0.557 |
| 6 | 0.656 | 0.556 |
| 8 | 0.656 | 0.556 |
| 10 | 0.654 | 0.554 |
| 12 | 0.646 | 0.546 |
| 14 | 0.648 | 0.538 |
| 16 | 0.647 | 0.537 |
| 18 | 0.637 | 0.527 |
| 20 | 0.627 | 0.487 |
| 22 | 0.591 | 0.461 |
| 24 | 0.567 | 0.437 |
| 26 | 0.548 | 0.408 |
| 28 | 0.527 | 0.407 |
| 30 | 0.506 | 0.406 |
| 32 | 0.503 | 0.403 |

Table S: Detailed performance comparison on TruthfulQA dataset with Phi-3-Mini.

| Layer | Phi-3-Mini MiniCache | Phi-3-Mini Mean |
|---|---|---|
| 0 | 19.686 | 19.465 |
| 2 | 19.385 | 19.365 |
| 4 | 19.368 | 19.221 |
| 6 | 19.100 | 18.255 |
| 8 | 19.038 | 17.019 |
| 10 | 19.500 | 15.912 |
| 12 | 19.216 | 15.525 |
| 14 | 20.026 | 15.195 |
| 16 | 19.641 | 15.058 |
| 18 | 18.756 | 14.763 |
| 20 | 17.738 | 14.593 |
| 22 | 19.546 | 14.182 |
| 24 | 19.723 | 13.954 |
| 26 | 18.366 | 13.919 |
| 28 | 18.402 | 12.231 |
| 30 | 16.827 | 12.158 |
| 32 | 16.635 | 10.333 |

