# OpenReview forum: "MiniCache: KV Cache Compression in Depth Dimension for Large Language Models"
_NeurIPS.cc/2024/Conference — NeurIPS 2024 poster_

### Official Review · Reviewer_GYW9 · 2024-06-30

**Soundness:** 2
**Presentation:** 2
**Contribution:** 3
**Rating:** 6
**Confidence:** 5

**Summary:**

This paper proposes a KV cache compression method by merging keys and values of consecutive layers. Based on the empirical observation that keys and values of consecutive layers after the mid-depth layer have high cosine similarity, the authors propose a merging strategy using angular interpolation. Additionally, they emphasize the importance of retention tokens based on their empirical findings. Extensive experiments across multiple state-of-the-art models and various evaluation benchmarks demonstrate the effectiveness of the proposed compression method.

**Strengths:**

- The proposed approach of merging keys and values can have a significant practical impact.
- The method has low computational overhead while providing meaningful efficiency improvements.
- The proposed approach is training-free.
- Experiments conducted on various models and benchmarks demonstrate the effectiveness of the proposed method.
- The writing is overall clear.

**Weaknesses:**

- Some methodological design choices seem ad-hoc. For example, the method starts merging after the mid-layer and only merges two consecutive layers. I believe the paper could be technically improved by addressing these design choices.
- The authors compare the efficiency of the method after applying quantization (MiniCache 4-bit) to the FP16 baseline. This comparison overstates the method's effectiveness because the contribution of the paper does not involve quantization. The paper should measure efficiency improvement without using quantization. Currently, lines L22-24, L302-303, etc., can mislead readers.
- A comparison to early layer-exiting approaches seems more relevant than comparisons to KV quantization methods, which the current draft focuses on.
   - e.g., "Fast and Robust Early-Exiting Framework for Autoregressive Language Models with Synchronized Parallel Decoding", EMNLP, 2023

**minor comments**
- Figure 1, index -> layer index
- Line 140, Figure 1 (b) -> Figure 1 (a)
- For related works, it would be informative to include token compression methods:
  - "Compressed Context Memory for Online Language Model Interaction", ICLR 2024
  - "Recurrent memory transformer", NeurIPS 2022

**Questions:**

- What will be the results when applying the merging function to more than two layers?
- How about applying the merging function based on cosine similarity? For instance, only apply merging when the cosine similarity is higher than a specified threshold. This might address the issues mentioned in the first weakness above.
- Could you compare the proposed method with early layer-exiting methods?
- In Eq (4), is the retention index defined per layer? If not, how do you measure d_i, d_{min}, and d_{max}?
- Could you provide an intuitive analysis of why some tokens have severely low cosine similarity in Figure 2 (b)? Are there any common characteristics of these tokens?
- Do you apply merging per attention head or to the entire key/value vectors?

**Limitations:**

In appendix.

---

> ### Author Rebuttal · Authors · 2024-08-07
>
> Thanks to the reviewer for the valuable comments.
>
> **Q1: Ad-hoc design choices (mid-layer merging, only two layers) need improvement.**
>
> **A1:**
> It is worth noting that MiniCache is the pioneer work to explore KV cache compression along the depth dimension (Lines 48-50). This insight has been highly recognized by Reviewer TyMG (*"...novel perspective of compressing across layers"*), Reviewer fj43 *(“The idea ... introduces a new perspective”)* and Reviewer ZDdv (*“This novel perspective, previously unexplored”*).
>
> With this new motivation, the design of MiniCache is initially based on strong empirical evidence as shown in Figure 1(a), where KV cache states exhibit high similarity between the adjacent layers in the middle-to-deep portion of LLMs. Thus,
>
> 1. Merging after the mid-layer is **a reasonable and good starting point for our early exploration** as it would ideally keep most performance of LLMs.
> 2. More importantly, **we have already shown that MiniCache can perform cache-merging across more layers in Figure 4**, e.g. from shallow to deep layers. We have also discussed cross-layer merging beyond two layers using an advanced merging algorithm termed Spherical Cubic Interpolation in further work (Lines 339-342).
>
> Furthermore, we agree that our current approach could be slightly simple as an early effort. However, at this stage, we have already conducted comprehensive experiments to demonstrate the effectiveness of our main idea of cross-layer KV cache merging. Notably, this has been **recognized by all reviewers.** Thus, we will leave more advanced design for future works, as also recognized by Reviewer fj43 ("could inspire further research in inter-layer redundancy exploitation.").
>
> **Q2: Results of merging more than two layers?**
>
> **A2:**
> We have considered multiple-layer merging functions; however, in the current version of MiniCache, we employ SLERP (Spherical Linear Interpolation) as our merging function. This choice is due to SLERP's inherent ability to merge only two vectors.
>
> To extend this capability to more than two layers, please refer to general response **Q3**.
>
> As discussed in Q1, we will explore merging across more than two layers in future work.
>
> **Q3: Merge based on cosine similarity with a threshold?**
>
> **A3:**
> In our approach, we used angular distance as the threshold for merging the KV cache.
> Mathematically, cosine similarity $\cos(\theta)$ can be converted to angular distance $d$ using the following relationship:
> $d = \frac{1}{\pi}\arccos(\cos(\theta))$
> Thus, merging based on angular distance is equivalent to merging based on cosine similarity.
>
>
>
> **Q4: Compare with early layer-exiting methods?**
>
> **A4:**
> Both early-exiting and MiniCache aim to accelerate inference by addressing the depth dimension. Early-exiting is more effective for compute-bound operations, whereas MiniCache is advantageous for memory-bound operations. Furthermore, early-exiting methods typically require extensive training, whereas MiniCache is training-free, making it broadly applicable to a wide range of off-the-shelf models and scenarios.
> In future work, we will explore combining these two paradigms. The relevant reference [C] will be included in the revised paper.
>
> **Q5: Is the retention index defined per layer in Eq (4)? How to measure d_i, d_{min}, and d_{max}?**
>
> **A5:**
> In Eq. (4), the retention index is **shared for the paired KV states by position between adjacent layers**. Specifically, $d_i$ represents the angular distance between the paired tokens, with $d_{min}$ and $d_{max}$ denoting the minimum and maximum distances across all paired tokens between two adjacent layers, respectively. The rationale behind this approach is to retain paired tokens when they exhibit low similarity, which corresponds to a large distance $d_i$ exceeding the retention threshold.
>
> **Q6: Why do some tokens have low cosine similarity in Fig 2(b)? Are there any common characteristics?**
>
> **A6:**
> We conducted a thorough analysis and observed that the salient tokens often include the initial token and punctuation, as illustrated in Figure I and Table IX of the rebuttal PDF.
>
> We have observed that the numerical values of these salient tokens are relatively large compared to other tokens. After passing through the attention layers, these salient tokens demonstrate significant numerical differences, as they receive strong attention despite not being semantically important. Similar phenomena have been observed in previous studies [D][E]. Consequently, the significant numerical differences between these tokens result in low similarity, rendering them unmergeable in our algorithm. We will include the analysis in the revised paper.
>
> **Q7: Merging per attention head or entire key/value vectors?**
>
> **A7:**
> We apply compression to the entire key and value vectors.
>
> **Q8: Efficiency comparison without quantization is needed to avoid misleading readers.**
>
> **A8:**
> Table VII in the rebuttal PDF demonstrates that MiniCache's cross-layer merging without quantization achieves almost lossless performance compared to the FP16 baseline, with a compression ratio of 1.53. In contrast, 2-bit quantization causes performance degradation. For instance, on the GSM8K dataset, performance drops from 0.159 to 0.127, but the compression ratio improves to 3.95.
>
> Our MiniCache method, focusing on the depth dimension, can complement any quantization and existing KV cache compression methods. The results indicate that combining cross-layer merging with 4-bit quantization achieves the optimal balance between performance and efficiency.
>
> To avoid confusion, we will revise the description as ``On the ShareGPT dataset, LLaMA-2-7B with cross-layer merging achieves a compression ratio of 1.53. Additionally, since MiniCache is orthogonal to existing quantization techniques, it can achieve a compression ratio of up to 5.02x when combined with the 4-bit quantization technique.”
>
> **Note: Please refer to the reference list in the general response.**

---

> ### Author Response · Authors · 2024-08-10
> **Follow-Up on Rebuttal**
>
> Dear Reviewer GYW9,
>
> We would like to thank you for your valuable feedback to improve our work. We are wondering whether our response has addressed your questions and can improve your opinion of our work. We are committed to ensuring that our work meets the expectations of our esteemed reviewers.
>
> Kindly let us know if you have any other concerns, and we will do our best to address them.
>
> Best regards,
> Authors of #1613

---

> > ### Comment · Reviewer_GYW9 · 2024-08-12
> > **Thank you for the rebuttal**
> >
> > Thank you for your detailed responses to my questions. I agree that the paper presents interesting observations. However, the approach still feels ad-hoc. It's unfortunate that no consideration was given to which layers should be merged. My suggestion in Q3 was to decide which layers to merge based on specific criteria. The authors only merge consecutive layer pairs after the mid-layer (please correct me if I’ve misunderstood). Regarding Q8, I believe a 1.5x improvement is indeed very significant, and I appreciate the authors for sharing this result. On the condition that the exaggerated claims in L21, 81, 94, etc. (regarding the proposed method achieving a 5x improvement) are revised accordingly, I raise my score to weak accept.

---

> > > ### Author Response · Authors · 2024-08-13
> > > **Thank you for the response**
> > >
> > > Dear Reviewer GYW9,
> > >
> > > Thank you for your valuable feedback and for raising the score. We are pleased that the new experiments and additional details have enhanced the clarity and positioning of our work.
> > >
> > > We understand the reviewer's concern regarding the criteria for layer selection. We will incorporate your suggestion and include a more detailed trade-off analysis based on the experimental results shown in Figure 4. This will also explore the potential for a more dynamic layer selection strategy in our future work.
> > >
> > > Best regards,
> > > Authors of #1613

---

### Official Review · Reviewer_ZDdv · 2024-07-08

**Soundness:** 4
**Presentation:** 3
**Contribution:** 4
**Rating:** 7
**Confidence:** 5

**Summary:**

Authors proposed a KVCache Compression Method, Minicache, is introduced as a method to efficiently compress the Key-Value (KV) cache in large language models (LLMs) by leveraging the high similarity of KV states between adjacent layers in the middle-to-deep portion of LLMs. This compression is achieved by disentangling states into magnitude and direction components and interpolating directions while retaining distinct state pairs to minimize storage overhead. MiniCache is training-free, complements existing compression strategies, and demonstrates significant performance improvements, including up to a 5.02x compression ratio, a 5x enhancement in inference throughput, and a 41% reduction in memory footprint, all while maintaining near-lossless performance across various models and benchmarks.

**Strengths:**

1.[Novel Approach, Interesting Observation] The authors explore KV cache compression from a novel perspective by focusing on the depth dimension, which is previously unexplored. They identify a significant finding: KV cache states exhibit high similarity between adjacent layers in the middle-to-later stages of LLMs. This discovery is backed by a thorough analysis and visualization of layer-wise similarity statistics, leading to a well-motivated methodology for compressing the KV cache. This approach has substantial implications for the development of KV cache compression techniques.

2.[Sound Methodology] The authors propose a training-free strategy for cross-layer merging, formulating a reparametrization compression technique to enhance inference efficiency. This method carefully considers outliers, retaining these specific tokens and restoring them through a well-constructed strategy.

3.[Strong Experiments, Well Compatibility] Extensive experiments across multiple datasets with four different models demonstrate that the proposed MiniCache consistently performs effective compression without compromising performance, achieving a superior compression ratio of ~5× and outperforming existing methods. Furthermore, the proposed method is compatible with existing quantization strategies, highlighting its practical values.

4.The paper is well-written, which is easy to understand.

**Weaknesses:**

1.I am concerned about the computational overhead during the reparametrization and restoration stages. Does this compression strategy increase computational overhead?
2.The authors selected the middle layers for cross-layer merging compression. Have the authors considered any principled methods to further increase the compression ratio by compressing the shallow layers in future work?

**Questions:**

See weaknesses.

**Limitations:**

The authors address the limitations inherited from the SLERP method and propose future research directions to enhance the adaptability of the merge compression strategy.

---

> ### Author Rebuttal · Authors · 2024-08-07
>
> Thanks to the reviewer for the valuable comments.
>
> **Q1 : Concern about computational overhead during reparametrization and restoration stages. Does this compression strategy increase computational overhead?**
>
> **A1:**
> Reparametrization-based compression involves computing magnitude and direction vectors, and applying the SLERP algorithm through simple matrix manipulation. Additionally, we compute pairwise distances to identify salient tokens, as detailed in Section 4.2.
>
> For restoration, we utilize an in-place index-based token replacement operation, where the computational overhead is managed by the PyTorch index selection procedure, as mentioned in line 216.
>
> As shown in Table V of the rebuttal PDF, the reparametrization process, including the computation of magnitude and direction, takes 0.093 ms (0.031 ms + 0.062 ms), and the restoration process, including the computation of the distance matrix and token replacement, takes 0.116 ms (0.061 ms + 0.055 ms). These times indicate a negligible computational overhead compared to the overall attention computation, which takes 9.756 ms. The running time is measured in milliseconds per attention layer with a batch size of 1 on LLaMA-2-7B.
>
> **Q2: The authors selected the middle layers for cross-layer merging compression. Have the authors considered any principled methods to further increase the compression ratio by compressing the shallow layers in future work?**
>
> **A2:**
> Yes, we have considered and proposed methods to further improve the compression ratio.
>
> Our observations indicate that certain shallow layers exhibit low similarities, suggesting the presence of differences and residual errors. To address this, we suggest employing a meta-learning-based approach to approximate these differences in the shallow layers. Once these approximations are made, we can bridge the gaps in the shallow layers, enabling their effective merging and compression. The possibility of merging shallow layers has been demonstrated in concurrent work [B], though it trains from scratch.
>
> References:
>
> [B] You only cache once: Decoder-decoder architectures for language models. Arxiv 2024.

---

> > ### Comment · Reviewer_ZDdv · 2024-08-12
> >
> > My concerns are fully addressed and I would like to keep my score.

---

### Official Review · Reviewer_fj43 · 2024-07-10

**Soundness:** 3
**Presentation:** 3
**Contribution:** 2
**Rating:** 6
**Confidence:** 4

**Summary:**

The paper proposes a novel method called MiniCache for compressing the KV cache in large language models (LLMs) by merging cache states across layers. The authors argue that this approach significantly reduces the memory footprint and enhances inference throughput without significant performance loss. The paper includes evaluations on various models and benchmarks, demonstrating the effectiveness of MiniCache in terms of compression ratio and throughput improvement.

**Strengths:**

1. The idea of compressing the KV cache by merging states across layers introduces a new perspective on reducing memory usage during inference. This approach could inspire further research in inter-layer redundancy exploitation.
2. Memory consumption and inference speed are critical challenges in deploying LLMs. The proposed MiniCache method addresses these issues directly, offering a potentially valuable tool for practitioners working with resource-constrained environments.
3. The method does not require additional training, making it easy to integrate into existing inference pipelines without the need for extensive retraining or fine-tuning of models.

**Weaknesses:**

**Major Weekness 1: Lack of Baseline Comparisons**
In Table 1, the authors only include quantization methods for performance comparison, neglecting other KV cache eviction methods such as those proposed in [1] and [2].

**Major Weekness 2: Lack of Evaluation for Instruction-following Benchmarks**
Given that instruction-tuned models are more generalizable for downstream applications, it is essential to evaluate how MiniCache performs on instruction-following benchmarks such as MT-Bench [3].


[1] H2O: Heavy-Hitter Oracle for Efficient Generative Inference of Large Language Models, NeurIPS 2023
[2] Efficient Streaming Language Models with Attention Sinks, ICLR 2024
[3] Judging LLM-as-a-Judge with MT-Bench and Chatbot Arena, NeurIPS 2023

**Questions:**

1. Beyond memory compression, what is the impact of MiniCache on latency?

**Limitations:**

As mentioned above.

---

> ### Author Rebuttal · Authors · 2024-08-07
>
> Thanks to the reviewer for the valuable comments.
>
> **Q1: Lack of Baseline Comparisons. In Table 1, the authors only include quantization methods for performance comparison, neglecting other KV cache eviction methods such as those proposed in [14] and [15].**
>
> **A1:** We have included the baseline comparison with H2O [14] in Appendix Section A, along with additional experimental results. We further add the Attention Sink [15] benchmark, as shown in Table III in the rebuttal PDF.
>
> According to this table, MiniCache consistently outperforms H2O [14] and Attention Sink [15]. Additionally, MiniCache explores the KV Cache in a **novel depth perspective**, making it **orthogonal to existing quantization and sparsity techniques** for further improvements.
>
> **Q2: Lack of Evaluation for Instruction-following Benchmarks. Given that instruction-tuned models are more generalizable for downstream applications, it is essential to evaluate how MiniCache performs on instruction-following benchmarks such as MT-Bench [A].**
>
> **A2:**
> We benchmark the MiniCache using the MT-Bench dataset, which focuses on instruction-following tasks, as shown in Table VIII in the rebuttal PDF. The models employed for this benchmarking include LLaMA-2-7B-Chat, Mistral-7B-Instruct, and LLaMA-3-8B-Instruct. Our MiniCache-4bit exhibits a reasonably slight performance reduction caused by 4-bit quantization. However, MiniCache can achieve almost lossless performance via standalone cross-layer merging.
>
> **Q3: Latency Benchmark. Beyond memory compression, what is the impact of MiniCache on latency?**
>
> **A3:**
> We benchmark the latency of LlaMA-2-7B on an NVIDIA A100 GPU using different sequence lengths ranging from 1024 to 4096 with a batch size of 16, as shown in Table IV in the rebuttal PDF. We compare it with H2O, which requires calculating full attention scores to estimate token importance. In contrast, MiniCache performs reparameterization-based merging and token restoration using simple matrix manipulations, resulting in more lightweight computations and lower latency. Specifically, when the sequence length is 4096, MiniCache shows a 36.83% reduction in latency compared to H2O.
>
> References:
>
> [A] Judging LLM-as-a-Judge with MT-Bench and Chatbot Arena, NeurIPS 2023.

---

### Official Review · Reviewer_TyMG · 2024-07-17

**Soundness:** 3
**Presentation:** 3
**Contribution:** 3
**Rating:** 7
**Confidence:** 4

**Summary:**

This paper introduces MiniCache, a novel approach to compressing the Key-Value (KV) cache in large language models (LLMs) to enhance inference efficiency. The KV cache is crucial in storing key-value states of previously generated tokens, significantly reducing redundant computations and lowering latency during autoregressive generation. However, as the sequence length increases, the KV cache's size also grows linearly, leading to substantial memory consumption. MiniCache addresses this issue by compressing the KV cache across layers from a depth perspective, leveraging the observation that KV cache states exhibit high similarity between adjacent layers in the middle-to-deep portions of LLMs. The proposed method involves disentangling the states into magnitude and direction components, interpolating the directions while preserving the lengths, and retaining highly distinct state pairs unmerged to minimize information loss. MiniCache is a training-free, general approach that complements existing KV cache compression strategies like quantization and sparsity.

The authors conducted comprehensive evaluations using various models, including LLaMA-2, LLaMA-3, Phi-3, Mistral, and Mixtral, across multiple benchmarks. The results demonstrated that MiniCache achieves superior compression ratios and high throughput, with LLaMA-2-7B showing a compression ratio of up to 5.02×, a 5× increase in inference throughput, and a 41% reduction in memory footprint compared to the FP16 full cache baseline, all while maintaining near-lossless performance. The paper highlights the potential of MiniCache to significantly reduce memory requirements and enhance the efficiency of LLM inference, making it a promising solution for applications requiring long context inputs and extensive sequence generation.

**Strengths:**

The paper presents a substantive contribution to the field of efficient machine learning by introducing a novel, high-quality, and clearly explained method for KV cache compression in large language models. The significance of the work is underscored by its potential to improve the practicality and scalability of LLMs, addressing a critical bottleneck in their deployment. The originality of the approach, combined with comprehensive evaluations and clear exposition, makes this paper a valuable addition to the literature on efficient ML techniques.

**Originality**

- **Novel Compression Approach**: The paper introduces a unique method for KV cache compression in large language models by exploring the depth dimension, which is a previously overlooked area. This novel perspective of compressing across layers rather than within layers demonstrates a creative combination of existing ideas in a new, impactful way.
- **Reparameterization Strategy**: The method's use of reparameterization to disentangle state vectors into magnitude and direction components for interpolation is innovative. This approach preserves important information while effectively reducing memory usage.

**Quality**

- **Comprehensive Evaluation**: The authors provide a thorough evaluation of MiniCache across various models and benchmarks, including LLaMA-2, LLaMA-3, Phi-3, Mistral, and Mixtral. The extensive experiments validate the method's effectiveness and robustness.
- **Performance Metrics**: The results show significant improvements in compression ratios, inference throughput, and memory footprint reduction, with metrics such as a 5.02× compression ratio and a 41% reduction in memory usage while maintaining near-lossless performance. These metrics highlight the quality and practicality of the proposed solution.

**Clarity**

- **Detailed Exposition**: The paper is well-written, with a clear and detailed exposition of the methodology. Figures and tables are effectively used to illustrate key concepts, observations, and results, aiding in the understanding of the approach and its benefits.
- **Step-by-Step Explanation**: The authors provide a step-by-step explanation of the MiniCache method, from the initial observations to the final implementation, making the paper accessible even to those less familiar with the intricacies of KV cache compression.

**Significance**

- **Addressing a Critical Issue**: The paper addresses a significant challenge in the deployment of large language models – the growing memory consumption of KV caches with increasing sequence lengths. By reducing the memory footprint and improving inference efficiency, MiniCache has the potential to make LLMs more practical and scalable in real-world applications.
- **Broad Applicability**: The approach is general and training-free, making it applicable to a wide range of models and scenarios. This broad applicability enhances the significance of the work, as it can be integrated into existing systems with minimal modification.

**Weaknesses:**

1. **No Implementation Source Code Provided**: The paper does not include the implementation source code, which is a significant limitation. Releasing the source code would facilitate further research and enable other researchers to replicate and build upon the work. Providing the code upon publication would enhance the paper's impact and encourage broader adoption of the proposed method.

2. **Insufficient Justification for SLERP**: The introduction of Spherical Linear Interpolation (SLERP) in the paper feels abrupt and lacks sufficient justification. While SLERP is used for interpolating between vectors, the paper does not provide enough rationale for why this specific technique was chosen over other interpolation methods. More ablation studies should be conducted to demonstrate the effectiveness and necessity of using SLERP in this context. These studies could compare SLERP with alternative interpolation techniques to show its advantages and validate the authors' choice.

### Minor Writing Improvements

1. **Figure Clarity**:
   - **Figure 1(a)**: The resolution of Figure 1(a) could be higher, or the figure could be replaced with a vector graphic to improve clarity and readability. Enhancing the visual quality would make the figure easier to understand and more professional.

2. **Typographical Corrections**:
   - **Line 308**: The word "interpretation" should be corrected to "interpolation".
   - **Line 328**: The phrase "A larger t" should be corrected to "A larger \(\gamma\)".

**Questions:**

1. **Distance-Based Threshold for Retention (Line 226)**
   Why did you choose a distance-based threshold instead of the merging ratio of overall tokens as the control for retention? Can you show the effect on accuracy and efficiency as the ratio of retention tokens varies, highlighting this trade-off?

2. **Ratio of Merged Tokens in Ablation Study**
   In the second ablation study, can you show the ratio of merged tokens and the efficiency trade-offs for different settings?

3. **Implementation Source Code**
   Can you include the implementation source code?

4. **Justification for SLERP**
   Can you justify your choice of SLERP over other interpolation methods?

**Limitations:**

There should be a section discussing the limitation of this work and its social impact.

---

> ### Author Rebuttal · Authors · 2024-08-07
>
> Thanks to the reviewer for the valuable comments.
>
> **Q1: Distance-Based Threshold for Retention (Line 226)**. **Why did you choose a distance-based threshold instead of the merging ratio of overall tokens as the control for retention? Can you show the effect on accuracy and efficiency as the ratio of retention tokens varies, highlighting this trade-off?**
>
> **A1:**
> According to our Observation 2, token pairs with lower similarity scores are more critical for restoration, and different adjacent layers exhibit distinct patterns.
>
> Our proposed **dynamic distance-based threshold** allows us to selectively retain salient tokens according to low similarity scores. Setting the retention ratio of the overall tokens as a hyperparameter is impractical because determining the optimal retention ratio for each layer manually is challenging due to the dynamic nature across different layers.
>
> To illustrate the trade-off regarding accuracy and efficiency, we conducted experiments by varying the ratio of retention tokens, as shown in Table I in the rebuttal PDF. We observe that as the overall token retention ratio increases, accuracy improves up to a certain point before plateauing. Retaining 20% of the tokens is necessary to ensure all salient tokens are preserved without performance degradation. In contrast, using a **dynamic distance-based threshold**, as proposed in our paper, we only need to retain the top 5% of the salient tokens. Note that efficiency decreases as more tokens are retained. This demonstrates that our distance-based approach better balances performance and efficiency than the fixed retention ratio counterpart.
>
> **Q2: Ratio of Merged Tokens in Ablation Study. In the second ablation study, can you show the ratio of merged tokens and the efficiency trade-offs for different settings?**
>
> **A2:**
> We are adding another column in terms of compression ratio to represent the efficiency trade-offs, as shown in Table II in the rebuttal PDF. The results suggest that setting γ to 0.05 achieves the best balance between performance and efficiency.
>
> **Q3: No Implementation Source Code Provided. Can you include the implementation source code?**
>
> **A3:**
> We will definitely release our source code upon acceptance.
>
> To ensure reproducibility, we have included comprehensive pseudocode in the Appendix, specifically in Algorithm 1 and Algorithm 2.
>
> - **Algorithm 1:** This algorithm outlines the overall logic of the MiniCache Inference process, covering both the prefilling and decoding stages. It provides a step-by-step breakdown of the core processes involved in our approach.
> - **Algorithm 2:** This algorithm details the compression and restoration processes. It elaborates on how the KV cache is compressed and subsequently restored, highlighting the technical intricacies and optimizations employed in our method.
>
> We believe that the inclusion, along with the detailed comments, will significantly aid the community in grasping the logic and mechanics of our implementation.
>
> **Q4: Justification for SLERP. Can you justify your choice of SLERP over other interpolation methods?**
>
> **A4:**
> 1. Motivation for Using SLERP:
> We conceptualize the KV Cache as activations, which consist of two factors: magnitude and direction in the spherical feature space. SLERP allows us to use a more compact form to represent the original KV Cache by merging states effectively. We can compute the overall magnitude vector and store it in a channel with a dimension equal to 1, significantly reducing the memory overhead, as shown in *section 4.2*.
>
> 2. Performance Metrics and Empirical Evidence:
> In our initial experiments, we considered average merging as the preliminary method; however, the performance in terms of accuracy was not promising, as shown in Figure 2(a). Subsequently, we conducted experiments using maximum norm-preserving interpolation, but it demonstrated lower accuracy compared to SLERP, as detailed in Appendix Section C (Line 579). Ultimately, we selected SLERP [64] as our final solution due to its multiple advantages in key performance metrics, including compression ratio, accuracy, and computational efficiency.
>
> To further substantiate our claim, we conducted additional ablation studies comparing SLERP with average merging and maximum norm-preserving interpolation. The results, as shown in Table VI in the rebuttal PDF, reaffirm that SLERP provides superior performance in information preservation.

---

> > ### Comment · Reviewer_TyMG · 2024-08-13
> > **Thank you for the response**
> >
> > I appreciate the authors' responses, and most of my concerns have been addressed. I will keep my evaluation for acceptance

---

### Author Rebuttal · Authors · 2024-08-07

## Response to all reviewers

We sincerely thank all reviewers for their valuable comments.

All reviewers agree that:

**The Novel Approach:**

- "The paper introduces a unique method …. This novel perspective impacts the field in a new, impactful way." (Reviewer TyMG)
- "The idea ... introduces a new perspective ... . This approach could inspire further research ..."(Reviewer fj43)
- "This novel perspective, previously unexplored, identifies a significant finding ..." (Reviewer ZDdv)

**The Insightful Observation:**

- "They identify a significant finding ... A thorough analysis and visualization back this discovery... ." (Reviewer ZDdv)
- "The authors provide a step-by-step explanation ... , from the initial observations to the final implementation." (Reviewer TyMG)

**The Sound Methodology:**

- "The method is innovative, preserving important information while effectively reducing memory usage." (Reviewer TyMG)
- "The authors propose a training-free strategy ... . This method ... a well-constructed strategy" (Reviewer ZDdv)
- "The method does not require additional training, making it easy to integrate into existing inference pipelines ..." (Reviewer fj43)

**The Practical Impact:**

- "MiniCache offers a potentially valuable tool." (Reviewer fj43)
- "The proposed approach has a significant practical impact." (Reviewer GYW9)
- "The approach is general and training-free ... have broad applicability ..." (Reviewer TyMG)

**The Comprehensive Experiments:**

- "The authors provide a thorough evaluation across various models and benchmarks. The extensive experiments validate the method's effectiveness and robustness." (Reviewer TyMG)
- "Extensive experiments demonstrate that MiniCache consistently performs effective compression without compromising performance." (Reviewer fj43)

## General Response

**Q1: Comparative Analysis and Benchmarks (Reviewers GYW9 and fj43)**

- We include comparisons with other KV cache eviction methods (e.g., those proposed in H2O and Attention Sinks) to strengthen our evaluation, as shown in Table III in the rebuttal PDF.
- We evaluate our method on instruction-following benchmarks like MT-Bench to demonstrate its generalizability for downstream applications, as shown in Table VIII in the rebuttal PDF.

**Q2: Efficiency and Overhead (Reviewers GYW9 and fj43)**

- We address concerns regarding computational overhead by measuring the different stages introduced during the reparametrization-based merging and token restoration, as shown in Table V in the rebuttal PDF.
- We evaluate the latency of MiniCache and compare latency with a baseline method H2O, as shown in Table IV in the rebuttal PDF. We will include these results in the revised paper.

**Q3: Future work and direction (Reviewers GYW9 and ZDdv)**

- Advanced Merging Techniques: Techniques such as Spherical Cubic Interpolation [75], mentioned in the further work Section 7, allow for the interpolation of multiple vectors. This method enables the effective merging of more than two layers.
- Multiple-Round Based Merging Strategy: This strategy involves iteratively merging layers by first combining two layers, then merging the resultant layer with a third one. However, this approach is less efficient and may introduce additional computational complexity.
- Approximation-Based Methods: To merge the KV cache in shallow layers, we plan to develop a meta-learning method to approximate their differences and merging errors. As shown in Figure 1 (a), some shallow layers exhibit low similarity, indicating significant differences. By approximating these differences, we can effectively merge and compress the KV cache in shallow layers, thereby enhancing the overall compression ratio.

Additionally, we thank all minor writing improvements given by Reviewers TyMG and GYW9. We will take these suggestions into account and improve them in the final manuscript.

**Reference:**

[A] Judging LLM-as-a-Judge with MT-Bench and Chatbot Arena. NeurIPS 2023.

[B] You only cache once: Decoder-decoder architectures for language models. Arxiv 2024.

[C] Fast and Robust Early-Exiting Framework for Autoregressive Language Models with Synchronized Parallel Decoding. EMNLP 2023

[D] Quantizable Transformers: Removing Outliers by Helping Attention Heads Do Nothing. NeurIPS 2023.

[E] Efficient Streaming Language Models with Attention Sinks. ICLR 2024.

---

### Decision · Program_Chairs · 2024-09-25

**Decision:**

Accept (poster)

**Comment:**

The AC has carefully read all reviewers' comments and the authors' response. The overall sentiment to the paper is quite positive. The main reasons for accepting the paper are:
1) The novel approach -- KV cache compression along the depth dimension (3 out of 4 reviewers).
2) Insightful observations -- KV cache at certain layers exhibit high similarity (2 out of 4 reviewers).
3) Sound and easy-to-us methodology (3 out of 4 reviewers).
4) Comprehensive evaluation -- across multiple models and benchmarks (2 out of 4 reviewers).

The main concerns from the reviewers are:
1) Lack of justification of design choices, e.g., SLERP.
2) Missing important baselines.
3) Ad-hoc design, e.g., layer selection policy.

Some of the concerns were addressed via the rebuttal, such as (1) and (2). The other is indeed weakness but can be further improved via follow-up work.

Overall, this is an interesting paper that explores a previous less looked dimension for compressing KV cache. It would be great if the authors can also add the actual memory usage, latency, throughput with batch size >1 comparison between MiniCache and alternative methods.